# WUSH: Near-Optimal Adaptive Transforms for LLM Quantization

Jiale Chen [1]   Vage Egiazarian [1]   Roberto L. Castro [2]   Torsten Hoefler [3]   Dan Alistarh [1 2]

## Abstract

Quantizing LLM weights and activations is a standard approach for efficient deployment, but a few extreme outliers can stretch the dynamic range and amplify low-bit quantization errors. Prior transform-based mitigations (e.g., Hadamard rotations) are fixed and data-agnostic, and their optimality for quantization has remained unclear. We derive closed-form optimal linear blockwise transforms for joint weight-activation quantization under standard RTN AbsMax-scaled block quantizers, covering both integer and floating-point formats. The resulting construction, WUSH, combines a Hadamard backbone with a data-dependent second-moment component to form a non-orthogonal transform that is provably near-optimal for FP and INT quantizers under mild assumptions while admitting an efficient fused GPU implementation. Empirically, WUSH improves W4A4 accuracy over the strongest Hadamard-based baselines (e.g., on Llama-3.1-8B-Instruct in MXFP4, it gains +2.8 average points with RTN and +0.7 with GPTQ) while delivering up to $5.8\times$ per-layer throughput over BF16 via FP4 MatMul. Source code is available at https://github.com/IST-DASLab/WUSH.

## 1. Introduction

Quantization of model weights (Dettmers et al., 2022; Frantar et al., 2023; Lin et al., 2024) or activations (Xiao et al., 2023; Ashkboos et al., 2024b) is now a standard tool for shrinking and accelerating large language models (LLMs) (Kurtic et al., 2025), making low-precision inference feasible on a wide range of hardware. A central difficulty, however, is that a few extreme "outlier" weights and activations expand the dynamic range and thereby degrade the effective resolution of low-bit representations.

[1]Institute of Science and Technology Austria (ISTA) [2]Red Hat AI [3]ETH Zürich. Correspondence to: Jiale Chen <jiale.chen@ist.ac.at>.

*Proceedings of the $43^{rd}$ International Conference on Machine Learning*, Seoul, South Korea. PMLR 306, 2026. Copyright 2026 by the author(s).

One way to mitigate such outliers is to apply linear transforms before quantization (Jegou et al., 2008; Jégou et al., 2011); in the case of LLM quantization, this is often in the form of rotations that spread variance more evenly across channels (Chee et al., 2023; Tseng et al., 2024a; Ashkboos et al., 2024b; Liu et al., 2025). For LLMs, Hadamard rotations have been remarkably effective, and blockwise variants aligned with quantization groups have also been shown to be useful in practice. Egiazarian et al. (2026) recently showed that the blockwise Hadamard transform offers the best empirical performance for quantization among a large set of existing transforms. Yet, these transforms are typically fixed and data-agnostic: in particular, the Hadamard transform does not adapt to the statistics of the underlying weights or activations. This raises a natural question: if the Hadamard transform is *not data-aware*, in what sense can it be considered optimal for quantization?

In this work, we address this question by deriving *closed-form optimal linear blockwise transforms* for joint weight-activation quantization, which are generally *non-orthogonal* and *adaptive*, i.e., data-aware. As opposed to prior methods such as SpinQuant (Liu et al., 2025) or FlatQuant (Sun et al., 2025) that learn transforms on calibration data via iterative optimizations (e.g., gradient descent), our approach is a closed-form calibration-driven transform. We call our construction *WUSH*, which is a mnemonic for its composition in Eq. (7). We show that WUSH is optimal for floating-point (FP) block quantizers and asymptotically optimal for integer (INT) block quantizers. Empirically, WUSH significantly improves end-to-end accuracy over prior transform-based baselines, substantially narrows the gap between NVFP and MXFP formats, and provides consistent benefits when combined with GPTQ (Frantar et al., 2023). It improves W4A4 (4-bit weights, 4-bit activations) accuracy by up to +2.8 average points (MXFP4 RTN) and +0.7 points (MXFP4 GPTQ) over the Hadamard-based baseline on Llama-3.1-8B-Instruct. Despite using a distinct (data-aware) transform per block, WUSH admits an efficient fused GPU kernel whose throughput matches that of optimized blockwise Hadamard kernels while delivering up to $5.8\times$ per-layer speedups over BF16 due to the lower-precision (FP4) matrix multiplication.

## 2. Related Work

**Post-training quantization (PTQ) and outliers.** PTQ schemes, such as RTN (round-to-nearest), OBQ (Frantar & Alistarh, 2022), and GPTQ (Frantar et al., 2023), are sensitive to heavy-tailed outliers in the weights and activations of LLMs because a small number of extreme values can determine the quantization scale, leading to high error.

**Non-uniform bitwidths and explicit outlier storage.** One common strategy to mitigate outliers is to use non-uniform bitwidths. Methods such as LLM.int8() (Dettmers et al., 2022), SpQR (Dettmers et al., 2024), and QUIK (Ashkboos et al., 2024a) explicitly separate and store outliers in higher precision, while HPTQ (Chen et al., 2026) uses Huffman encoding to compress the outliers and reduce their storage costs. These approaches can achieve high accuracy at a low effective bitwidth; however, the resulting formats are irregular and require specialized kernels.

**Transform-based outlier mitigation.** Another line of work applies transforms to the weights and activations before quantization to reduce the impact of outliers. SmoothQuant (Xiao et al., 2023) and AWQ (Lin et al., 2024) rescale channels to stabilize the dynamic ranges of weights and activations. QuIP (Chee et al., 2023) introduces an incoherence processing step. QuIP# (Tseng et al., 2024a), QTIP (Tseng et al., 2024b), and QuaRot (Ashkboos et al., 2024b) use Hadamard transforms to spread outlier energy across channel dimensions, while SpinQuant (Liu et al., 2025) and FlatQuant (Sun et al., 2025) learn transforms optimized on calibration data. However, their transforms are heuristic and costly to learn, which limits their accuracy or practicality when applied to fast, per-token activation quantization in large models.

**Blockwise transforms and emerging FP formats.** Recent FP formats, such as MXFP (MX Alliance, 2023) and NVFP (NVIDIA, 2024), have motivated blockwise transform schemes. Shao et al. (2025) indicates that blockwise Hadamard transforms can be more effective under these FP formats than full-layer transforms, but Egiazarian et al. (2026) highlights the limited gains achievable with Hadamard alone, linking this to the properties of the underlying weight and activation distributions. In contrast, this work (WUSH) derives a closed-form, data-aware optimal blockwise transform and analyzes how such a transform interacts with both FP and INT block quantizers.

## 3. Methodology

**Problem setup.** We formulate the transformed weight-activation quantization problem as follows. Define $\boldsymbol{W} \in \mathbb{R}^{d_{\mathrm{in}} \times d_{\mathrm{out}}}$ as the weight of a linear layer with $d_{\mathrm{in}}$ input channels and $d_{\mathrm{out}}$ output channels. Define $\boldsymbol{X} \in \mathbb{R}^{d_{\mathrm{in}} \times d_{\mathrm{batch}}}$ as the calibration input activation with $d_{\mathrm{in}}$ embedding chan-

nels and $d_{\mathrm{batch}}$ tokens. Define $q\left(\cdot\right)$ as a quantizer that maps a matrix of continuous values to quantized values (we will specify $q$ later). Let $\boldsymbol{T}_{\mathrm{W}}, \boldsymbol{T}_{\mathrm{X}} \in \mathbb{R}^{d_{\mathrm{in}} \times d_{\mathrm{in}}}$ be the transforms applied to $\boldsymbol{W}$ and $\boldsymbol{X}$, respectively. For weight-activation quantization, the output activation $\boldsymbol{W}^{\top} \boldsymbol{X}$ becomes $q\left(\boldsymbol{T}_{\mathrm{W}} \boldsymbol{W}\right)^{\top} q\left(\boldsymbol{T}_{\mathrm{X}} \boldsymbol{X}\right)$. Our goal is to choose $\boldsymbol{T}_{\mathrm{W}}$ and $\boldsymbol{T}_{\mathrm{X}}$ such that the $L_2$ output loss

$$\ell = d_{\mathrm{out}}^{-1} d_{\mathrm{batch}}^{-1} \left\| q\left(\boldsymbol{T}_{\mathrm{W}} \boldsymbol{W}\right)^{\top} q\left(\boldsymbol{T}_{\mathrm{X}} \boldsymbol{X}\right) - \boldsymbol{W}^{\top} \boldsymbol{X} \right\|_{\mathrm{F}}^{2} \quad (1)$$

is minimized under the given quantizer $q$. The constant $d_{\mathrm{out}}^{-1} d_{\mathrm{batch}}^{-1}$ is added to simplify the analysis later.

**Block-independent constraint.** While the weights can be pre-quantized, the activations must be transformed and quantized dynamically at inference time. To reduce the computational overhead of these operations, it is common (Egiazarian et al., 2026) to constrain $\boldsymbol{T}_{\mathrm{W}}, \boldsymbol{T}_{\mathrm{X}}$ to be block-diagonal transforms, and $q$ to be an RTN (round-to-nearest) quantizer with AbsMax (the maximum absolute value within a group) scales. We assume that the quantization is applied to each sub-column vector of shape $d \times 1$ and that the transform block size aligns with the quantization group size $d$, with $d$ being a factor of $d_{\mathrm{in}}$ and a power of 2. Denote the block partitions of the matrices as

$$
\begin{aligned}
\boldsymbol{T}_{\mathrm{W}} &= \mathrm{diag}\left(\boldsymbol{T}_{\mathrm{W}_{(1)}}, \ldots, \boldsymbol{T}_{\mathrm{W}_{(d_{\mathrm{in}}/d)}}\right), & \boldsymbol{T}_{\mathrm{W}_{(i)}} &\in \mathbb{R}^{d \times d}, \\
\boldsymbol{T}_{\mathrm{X}} &= \mathrm{diag}\left(\boldsymbol{T}_{\mathrm{X}_{(1)}}, \ldots, \boldsymbol{T}_{\mathrm{X}_{(d_{\mathrm{in}}/d)}}\right), & \boldsymbol{T}_{\mathrm{X}_{(i)}} &\in \mathbb{R}^{d \times d}, \\
\boldsymbol{W}^{\top} &= \left[\boldsymbol{W}_{(1)}^{\top}, \ldots, \boldsymbol{W}_{(d_{\mathrm{in}}/d)}^{\top}\right], & \boldsymbol{W}_{(i)} &\in \mathbb{R}^{d \times d_{\mathrm{out}}}, \\
\boldsymbol{X}^{\top} &= \left[\boldsymbol{X}_{(1)}^{\top}, \ldots, \boldsymbol{X}_{(d_{\mathrm{in}}/d)}^{\top}\right], & \boldsymbol{X}_{(i)} &\in \mathbb{R}^{d \times d_{\mathrm{batch}}}.
\end{aligned}
\quad (2)
$$

Then, we can express the output without and with weight-activation quantization, respectively, as

$$
\begin{aligned}
\boldsymbol{W}^{\top} \boldsymbol{X} &= \sum_{i=1}^{d_{\mathrm{in}}/d} \boldsymbol{W}_{(i)}^{\top} \boldsymbol{X}_{(i)}, \\
q(\boldsymbol{T}_{\mathrm{W}} \boldsymbol{W})^{\top} q(\boldsymbol{T}_{\mathrm{X}} \boldsymbol{X}) &= \sum_{i=1}^{d_{\mathrm{in}}/d} q\left(\boldsymbol{T}_{\mathrm{W}_{(i)}} \boldsymbol{W}_{(i)}\right)^{\top} q\left(\boldsymbol{T}_{\mathrm{X}_{(i)}} \boldsymbol{X}_{(i)}\right).
\end{aligned}
\quad (3)
$$

Finally, define the (normalized) blockwise output loss as

$$\ell_{(i)} = d_{\mathrm{out}}^{-1} d_{\mathrm{batch}}^{-1} \left\| q\left(\boldsymbol{T}_{\mathrm{W}_{(i)}} \boldsymbol{W}_{(i)}\right)^{\top} q\left(\boldsymbol{T}_{\mathrm{X}_{(i)}} \boldsymbol{X}_{(i)}\right) - \boldsymbol{W}_{(i)}^{\top} \boldsymbol{X}_{(i)} \right\|_{\mathrm{F}}^{2}. \quad (4)$$

We will approximate the layerwise loss as $\ell \approx \sum_{i=1}^{d_{\mathrm{in}}/d} \ell_{(i)}$ and minimize the blockwise losses $\ell_{(i)}$ independently.

**Optimal block-diagonal transforms.** In the following, we will prove, under reasonable assumptions, that there exist closed-form optimal transforms for each block. These transforms are provably optimal for FP and asymptotically

optimal for INT quantization. Let $\boldsymbol{W}'_{(i)}, \boldsymbol{X}'_{(i)} \in \mathbb{R}^{d \times d}$ be matrices that satisfy

$$
\begin{aligned}
\boldsymbol{W}'_{(i)} \boldsymbol{W}'^{\top}_{(i)} &= d_{\text{out}}^{-1} \boldsymbol{W}_{(i)} \boldsymbol{W}^{\top}_{(i)}, \\
\boldsymbol{X}'_{(i)} \boldsymbol{X}'^{\top}_{(i)} &= d_{\text{batch}}^{-1} \boldsymbol{X}_{(i)} \boldsymbol{X}^{\top}_{(i)},
\end{aligned} \tag{5}
$$

respectively. Without loss of generality (Appendix Section A.1), we let $\boldsymbol{W}'$ and $\boldsymbol{X}'$ be the lower triangular matrices from the Cholesky decomposition of the second moments $d_{\text{out}}^{-1} \boldsymbol{W}_{(i)} \boldsymbol{W}^{\top}_{(i)}$ and $d_{\text{batch}}^{-1} \boldsymbol{X}_{(i)} \boldsymbol{X}^{\top}_{(i)}$. In the case of $\text{rank}\left(\boldsymbol{W}_{(i)}\right) < d$ or $\text{rank}\left(\boldsymbol{X}_{(i)}\right) < d$, we can dampen the diagonal of the second moments before the Cholesky decomposition. Let the orthogonal matrices $\boldsymbol{U}_{(i)}, \boldsymbol{V}_{(i)} \in \mathbb{R}^{d \times d}$ and the diagonal matrix $\boldsymbol{S}_{(i)} \in \mathbb{R}^{d \times d}$ be the singular value decomposition (SVD) of

$$
\boldsymbol{W}'^{\top}_{(i)} \boldsymbol{X}'_{(i)} = \boldsymbol{U}_{(i)} \boldsymbol{S}_{(i)} \boldsymbol{V}^{\top}_{(i)}. \tag{6}
$$

Let $\boldsymbol{H} \in \left\{\pm d^{-\frac{1}{2}}\right\}^{d \times d}$ be a normalized (orthogonal) Hadamard matrix. Then, we will show that the optimal $\boldsymbol{T}_{\text{W}_{(i)}}$ and $\boldsymbol{T}_{\text{X}_{(i)}}$ can be constructed as

$$
\begin{aligned}
\boldsymbol{T}_{\text{xvsh}(i)} &= \boldsymbol{H} \boldsymbol{S}_{(i)}^{-\frac{1}{2}} \boldsymbol{V}^{\top}_{(i)} \boldsymbol{X}'^{\top}_{(i)}, \\
\boldsymbol{T}_{\text{wush}(i)} &= \boldsymbol{H} \boldsymbol{S}_{(i)}^{-\frac{1}{2}} \boldsymbol{U}^{\top}_{(i)} \boldsymbol{W}'^{\top}_{(i)},
\end{aligned} \tag{7}
$$

respectively. Note that[1]

$$
\boldsymbol{T}_{\text{xvsh}(i)} = \boldsymbol{T}_{\text{wush}(i)}^{-\top}, \tag{8}
$$

which can be easily verified by calculating $\boldsymbol{T}_{\text{xvsh}(i)} \boldsymbol{T}^{\top}_{\text{wush}(i)} = \mathbf{I}$, where $\mathbf{I}$ is the identity matrix. The WUSH construction also jointly balances the weight and activation scales. Appendix Section A.3 discusses and visualizes this effect.

**Remark.** The Hadamard matrix $\boldsymbol{H}$ is the only data-agnostic ingredient in our optimal formulation in Eq. (7). Incidentally, this explains why it has been empirically observed to be an effective data-agnostic orthogonal transform.

**GPTQ integration.** In practice, the second-order information $d_{\text{batch}}^{-1} \boldsymbol{X}_{(i)} \boldsymbol{X}^{\top}_{(i)}$ in Eq. (5) can be obtained either from the calibration activations or from the Hessian matrix; the latter coincides with the Hessian used in GPTQ. Therefore, it is a natural idea to integrate WUSH into GPTQ. However, GPTQ updates the weights iteratively via error propagation, while the WUSH transform is constructed from the second-order statistics of the updated weights. This coupling between weight updates and transform construction requires an interleaved computational schedule. Algorithm 2 summarizes the procedure used to compute the blockwise WUSH transforms and pre-quantized weights for a linear layer, using either RTN or GPTQ. The GPTQ error propagations are

[1]The $-\top$ superscript means $(\cdot)^{-\top} = \left((\cdot)^{-1}\right)^{\top} = \left((\cdot)^{\top}\right)^{-1}$.

---

**Algorithm 1** Compute WUSH for One Block

**Input:** weight second moment $\boldsymbol{M}_{\text{W}} \in \mathbb{R}^{d \times d}$, activation second moment $\boldsymbol{M}_{\text{X}} \in \mathbb{R}^{d \times d}$, damping ratio $\lambda \in \mathbb{R}_{\geq 0}$
**Output:** WUSH transform $\boldsymbol{T}_{\text{wush}} \in \mathbb{R}^{d \times d}$

---

Initialize $d \times d$ identity matrix $\mathbf{I}$
Initialize normalized $d \times d$ Hadamard matrix $\boldsymbol{H}$
$\boldsymbol{W}' \boldsymbol{W}'^{\top} \leftarrow \text{Cholesky}\left(\boldsymbol{M}_{\text{W}} + \lambda d^{-1} \text{tr}\left(\boldsymbol{M}_{\text{W}}\right) \mathbf{I}\right)$
$\boldsymbol{X}' \boldsymbol{X}'^{\top} \leftarrow \text{Cholesky}\left(\boldsymbol{M}_{\text{X}} + \lambda d^{-1} \text{tr}\left(\boldsymbol{M}_{\text{X}}\right) \mathbf{I}\right)$
$\boldsymbol{U} \boldsymbol{S} \boldsymbol{V}^{\top} \leftarrow \text{SVD}\left(\boldsymbol{W}'^{\top} \boldsymbol{X}'\right)$   ▷ Eq. (6)
$\boldsymbol{T}_{\text{wush}} \leftarrow \boldsymbol{H} \boldsymbol{S}^{-\frac{1}{2}} \boldsymbol{U}^{\top} \boldsymbol{W}'^{\top}$   ▷ Eq. (7)

---

**Algorithm 2** Compute WUSH and Pre-Quantize Weights

**Input:** weights $\boldsymbol{W} \in \mathbb{R}^{d_{\text{in}} \times d_{\text{out}}}$, activations $\boldsymbol{X} \in \mathbb{R}^{d_{\text{in}} \times d_{\text{batch}}}$, damping ratio $\lambda \in \mathbb{R}_{\geq 0}$
**Output:** WUSH transform $\boldsymbol{T}_{\text{wush}(i)} \in \mathbb{R}^{d \times d}$ and pre-quantized weight $\widetilde{\boldsymbol{W}}_{(i)} \in \mathbb{R}^{d \times d_{\text{out}}}$ for each block $i \in \{1, \ldots, d_{\text{in}}/d\}$

---

▷ RTN (round-to-nearest) case
**for** $i \leftarrow 1$ **to** $d_{\text{in}}/d$ (in parallel) **do**
  $\boldsymbol{M}_{\text{W}(i)}, \boldsymbol{M}_{\text{X}(i)} \leftarrow d_{\text{out}}^{-1} \boldsymbol{W}_{(i)} \boldsymbol{W}^{\top}_{(i)}, \; d_{\text{batch}}^{-1} \boldsymbol{X}_{(i)} \boldsymbol{X}^{\top}_{(i)}$
  $\boldsymbol{T}_{\text{wush}(i)} \leftarrow \text{WUSH}\left(\boldsymbol{M}_{\text{W}(i)}, \boldsymbol{M}_{\text{X}(i)}, \lambda\right)$ ▷ Algorithm 1
  $\boldsymbol{T}_{\text{xvsh}(i)} \leftarrow \boldsymbol{T}_{\text{wush}(i)}^{-\top}$   ▷ Eq. (8)
  $\widetilde{\boldsymbol{W}}_{(i)} \leftarrow q\left(\boldsymbol{T}_{\text{xvsh}(i)} \boldsymbol{W}_{(i)}\right)$   ▷ standard RTN
**end for**

---

▷ GPTQ case
Initialize $d_{\text{in}} \times d_{\text{in}}$ identity matrix $\mathbf{I}$
$\boldsymbol{\mathcal{H}} \leftarrow d_{\text{batch}}^{-1} \boldsymbol{X} \boldsymbol{X}^{\top}$   ▷ GPTQ's Hessian
$\left\{\boldsymbol{M}_{\text{X}(i)} \big| i \in \{1, \ldots, d_{\text{in}}/d\}\right\} \leftarrow$ diagonal blocks of $\boldsymbol{\mathcal{H}}$
    ▷ second moment of activations (same as the RTN case)
$\boldsymbol{L} \boldsymbol{L}^{\top} \leftarrow \text{Cholesky}\left(\left(\boldsymbol{\mathcal{H}} + \lambda d_{\text{in}}^{-1} \text{tr}\left(\boldsymbol{\mathcal{H}}\right) \mathbf{I}\right)^{-1}\right)$
    ▷ Cholesky of Hessian inverse (same as GPTQ)
$\left\{\boldsymbol{L}_{(i,j)} \in \mathbb{R}^{d \times d} \big| i, j \in \{1, \ldots, d_{\text{in}}/d\}\right\} \leftarrow$ blocks of $\boldsymbol{L}$
**for** $i \leftarrow 1$ **to** $d_{\text{in}}/d$ **do**
  $\boldsymbol{M}_{\text{W}(i)} \leftarrow d_{\text{out}}^{-1} \boldsymbol{W}_{(i)} \boldsymbol{W}^{\top}_{(i)}$
    ▷ second moment of updated weight block
  $\boldsymbol{T}_{\text{wush}(i)} \leftarrow \text{WUSH}\left(\boldsymbol{M}_{\text{W}(i)}, \boldsymbol{M}_{\text{X}(i)}, \lambda\right)$ ▷ Algorithm 1
  $\boldsymbol{T}_{\text{xvsh}(i)} \leftarrow \boldsymbol{T}_{\text{wush}(i)}^{-\top}$   ▷ Eq. (8)
  $\overline{\boldsymbol{W}}_{(i)} \leftarrow \boldsymbol{T}_{\text{xvsh}(i)} \boldsymbol{W}_{(i)}$   ▷ transformed weights
  $\overline{\boldsymbol{\mathcal{H}}}_{(i)} \leftarrow \boldsymbol{T}_{\text{wush}(i)} \left(\boldsymbol{L}_{(i,i)} \boldsymbol{L}^{\top}_{(i,i)}\right)^{-1} \boldsymbol{T}^{\top}_{\text{wush}(i)}$
    ▷ transformed Hessian
  $\widetilde{\boldsymbol{W}}_{(i)} \leftarrow \text{GPTQ}\left(\text{weight} = \overline{\boldsymbol{W}}_{(i)}, \text{Hessian} = \overline{\boldsymbol{\mathcal{H}}}_{(i)}\right)$
  ▷ standard GPTQ subroutine (intra-block error propagation)
  $\boldsymbol{E}_{(i)} \leftarrow \boldsymbol{T}_{\text{xvsh}(i)}^{-1} \widetilde{\boldsymbol{W}}_{(i)} - \boldsymbol{W}_{(i)}$   ▷ error in original space
  **for** $j \leftarrow i$ **to** $d_{\text{in}}/d$ (in parallel) **do**
    $\boldsymbol{W}_{(j)} \leftarrow \boldsymbol{W}_{(j)} + \boldsymbol{L}_{(j,i)} \boldsymbol{L}^{-1}_{(i,i)} \boldsymbol{E}_{(i)}$
    ▷ inter-block GPTQ error propagation
  **end for**
**end for**

decomposed into intra-block and inter-block updates similar to its original paper (Frantar et al., 2023). For the intra-block updates, we apply the standard GPTQ subroutine to the current transformed weight block using the corresponding transformed Hessian. For inter-block propagation, we follow GPTQ's usual blockwise updates. The model-level calibration pipeline closely follows that of GPTQ: layers are processed sequentially, and after processing a layer, the calibration activations are propagated through the quantized layer to provide the inputs for calibrating the next one. During inference, the forward pass of a linear layer is calculated as

$$\sum_{i=1}^{d_{\text{in}}/d} \widetilde{\boldsymbol{W}}_{(i)}^{\top} q\left(\boldsymbol{T}_{\text{wush}(i)} \widetilde{\boldsymbol{X}}_{(i)}\right) \tag{9}$$

with $\widetilde{\boldsymbol{W}}_{(i)} \in \mathbb{R}^{d \times d_{\text{out}}}$ being the pre-quantized transformed weight blocks and $\widetilde{\boldsymbol{X}}_{(i)} \in \mathbb{R}^{d \times d_{\text{batch}}}$ being the new activation blocks instead of the calibration ones.

**Complexity analysis for offline preprocessing.** The offline processing cost of WUSH is very similar to that of the standard GPTQ algorithm because WUSH and GPTQ require the same type of activation Hessian information, and the cost is dominated by forwarding the calibration activations through the model. In particular, when combining WUSH and GPTQ, WUSH only adds a negligible overhead on top of GPTQ. For a layer, WUSH processes only $d_{\text{in}}/d$ diagonal blocks. In a typical setting, $d \ll d_{\text{in}} \ll d_{\text{batch}}$ and $d \ll d_{\text{out}} \ll d_{\text{batch}}$. Per block, forming second moments costs $O\left(d^2 d_{\text{batch}}\right)$ time and $O\left(d d_{\text{batch}}\right)$ memory, and applying the transform to the weights (for pre-quantization) costs $O\left(d^2 d_{\text{out}}\right)$ time and $O\left(d d_{\text{out}}\right)$ memory, while the Cholesky/SVD and the multiplication steps on small $d \times d$ matrices cost only $O\left(d^3\right)$ time and $O\left(d^2\right)$ memory. Therefore, the total per-block offline time and memory costs are $O\left(d^2 d_{\text{batch}}\right)$ and $O\left(d d_{\text{batch}}\right)$, and the per-layer costs are $O\left(d d_{\text{in}} d_{\text{batch}}\right)$ and $O\left(d_{\text{in}} d_{\text{batch}}\right)$. As a comparison, computing the Hessian matrix of the standard GPTQ for a layer requires $O\left(d_{\text{in}}^2 d_{\text{batch}}\right)$ time and $O\left(d_{\text{in}} d_{\text{batch}}\right)$ memory.

**Costs for online inference.** The only additional inference-time overhead is the activation-side block transform consisting of $d \times d \times d_{\text{in}}/d = d d_{\text{in}}$ elements per layer, while the weight-side transform is absorbed into the pre-quantized weights. With 4-bit weights (plus groupwise 8-bit scales) and 16-bit transforms, this is roughly a $\frac{16 d d_{\text{in}}}{4 d_{\text{in}} d_{\text{out}}} = \frac{4d}{d_{\text{out}}}$ relative storage overhead per layer. The memory is dominated by the KV cache and model weights, and the storage cost of the transforms is negligible for typical LLM dimensions. The online transform is fused with activation quantization in a kernel detailed in Section 5, leading to negligible runtime overhead.

## 4. Theoretical Derivation

In this section, we focus on the question of finding optimal transforms that minimize the loss of one block $\ell_{(i)}$ in Eq. (4). To build intuition for the proof, Figure 1 visualizes how different transforms reshape the expected quantization error under FP and INT block quantizers, in a 2D toy setting. Throughout, we omit the block subscript $(i)$ to simplify notation. Due to space limitations, the detailed derivation steps of some equations are deferred to Appendix Section A.4.

### 4.1. Problem Setup and General Proof Approach

**Probabilistic reformulation.** We reformulate the problem from a probabilistic perspective. We split the matrices $\boldsymbol{W} = [\boldsymbol{w}_1, \ldots, \boldsymbol{w}_{d_{\text{out}}}]$ and $\boldsymbol{X} = [\boldsymbol{x}_1, \ldots, \boldsymbol{x}_{d_{\text{batch}}}]$ into columns $\boldsymbol{w}_k, \boldsymbol{x}_k \in \mathbb{R}^d$ and treat the columns as i.i.d. samples from $d$-dimensional independent distributions $\boldsymbol{w} \sim \mathcal{D}_{\text{w}}$ and $\boldsymbol{x} \sim \mathcal{D}_{\text{x}}$, respectively. We motivate this multivariate modeling choice in Appendix Section A.2. We can reinterpret the blockwise loss in Eq. (4) as

$$\ell = \mathbb{E}_{\boldsymbol{w},\boldsymbol{x}} \left( q\left(\boldsymbol{T}_{\text{W}} \boldsymbol{w}\right)^{\top} q\left(\boldsymbol{T}_{\text{X}} \boldsymbol{x}\right) - \boldsymbol{w}^{\top} \boldsymbol{x}\right)^2. \tag{10}$$

Similarly, we can also reinterpret Eq. (5) so that the matrices $\boldsymbol{W}'$ and $\boldsymbol{X}'$ satisfy

$$\begin{aligned} \boldsymbol{W}' \boldsymbol{W}'^{\top} &= \mathbb{E}_{\boldsymbol{w}} \boldsymbol{w} \boldsymbol{w}^{\top}, \\ \boldsymbol{X}' \boldsymbol{X}'^{\top} &= \mathbb{E}_{\boldsymbol{x}} \boldsymbol{x} \boldsymbol{x}^{\top}, \end{aligned} \tag{11}$$

respectively.

**Unbiased quantization.** We assume that $q$ is a stochastic and unbiased quantizer, i.e., for a vector $\boldsymbol{\alpha} \in \mathbb{R}^d$, the quantization error $\varepsilon\left(\boldsymbol{\alpha}\right) = q\left(\boldsymbol{\alpha}\right) - \boldsymbol{\alpha}$ is a random vector with $\mathbb{E}_{\varepsilon(\boldsymbol{\alpha})} \varepsilon\left(\boldsymbol{\alpha}\right) = \boldsymbol{0}$. We further constrain

$$\boldsymbol{T}_{\text{W}} = \boldsymbol{T}_{\text{X}}^{-\top} \tag{12}$$

such that $q\left(\boldsymbol{T}_{\text{W}} \boldsymbol{w}\right)^{\top} q\left(\boldsymbol{T}_{\text{X}} \boldsymbol{x}\right)$ is unbiased with respect to $\boldsymbol{w}^{\top} \boldsymbol{x}$. Then, using the first-order approximation, we can split $\ell$ in Eq. (10) into two non-negative terms:

$$\begin{aligned} \ell = &\left(\mathbb{E}_{\boldsymbol{w},\varepsilon(\boldsymbol{T}_{\text{W}} \boldsymbol{w})} \left\| \boldsymbol{X}'^{\top} \boldsymbol{T}_{\text{W}}^{-1} \varepsilon\left(\boldsymbol{T}_{\text{W}} \boldsymbol{w}\right) \right\|^2\right) \\ &+ \left(\mathbb{E}_{\boldsymbol{x},\varepsilon(\boldsymbol{T}_{\text{X}} \boldsymbol{x})} \left\| \boldsymbol{W}'^{\top} \boldsymbol{T}_{\text{X}}^{-1} \varepsilon\left(\boldsymbol{T}_{\text{X}} \boldsymbol{x}\right) \right\|^2\right). \end{aligned} \tag{13}$$

The detailed derivation is given in Appendix Eq. (38).

**Problem reduction.** We can compute the optimal $\boldsymbol{T}_{\text{W}}$ and $\boldsymbol{T}_{\text{X}}$ individually for each of the non-negative terms and verify that $\boldsymbol{T}_{\text{W}} = \boldsymbol{T}_{\text{X}}^{-\top}$. The two terms have similar forms and can be optimized in similar ways. We focus on finding the optimal $\boldsymbol{T}_{\text{X}}$ for the second term. Define the $d$-dimensional random variable $\boldsymbol{y} = \boldsymbol{W}'^{\top} \boldsymbol{x}$ and denote its distribution as

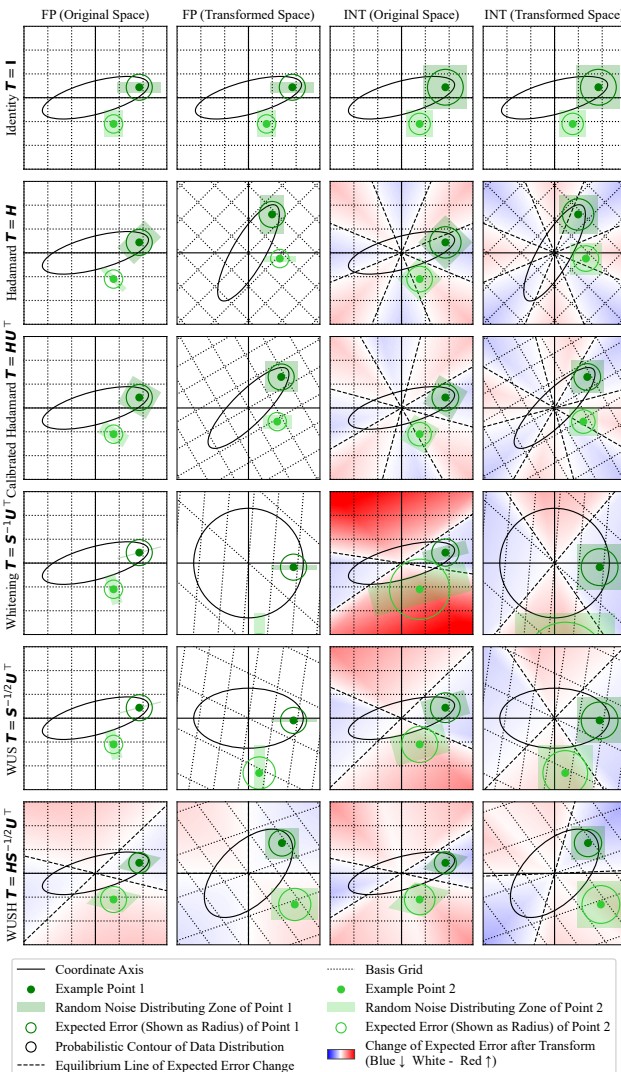

*Figure 1.* **2D illustration of how transforms shape the one-sided quantization error** $\left\|T^{-1}\varepsilon\left(Ty\right)\right\|^2$ **in Eq. (14) under FP and INT block quantizers.** Rows correspond to different transforms $T$ (identity; Hadamard: 45° rotation; calibrated Hadamard: equally spreading energy into each dimension; whitening; WUS: WUSH without Hadamard; WUSH), while columns show FP and INT quantizations in both original and transformed spaces. The dotted grid shows the basis induced by the transform. Two example points are shown in detail with their induced quantization noise support (Sections 4.2.1 and 4.3.1) represented as shaded parallelograms, and the corresponding average (expected) error magnitude (Eq. (14)) represented as the radii of circles (not necessarily proportional to the area of parallelograms). For each point on the plane, we visualize its expected error change (compared to that of the identity transform) calculated in the original space, with blue/red background heatmaps indicating lower/higher and dashed lines indicating the equilibrium set where the expected error is unchanged. No transform can reduce the error for all points on the plane. Thus, the key idea is to reduce the error around typical data distributions. The black ellipse depicts a representative data distribution contour. Overall, WUSH more effectively reduces the error by aligning the ellipse's major axis to the error reduction (blue) regions.

$y \sim \mathcal{D}_{\mathrm{y}}$. Define the transformation matrix $T = T_{\mathrm{X}} W'^{-\top}$. Then, the second term in Eq. (13) becomes

$$
\mathbb{E}_{x,\varepsilon(T_{\mathrm{X}}x)} \left\| W'^{\top} T_{\mathrm{X}}^{-1} \varepsilon\left(T_{\mathrm{X}}x\right) \right\|^2
$$
$$
= \mathbb{E}_{x,\varepsilon(T_{\mathrm{X}}x)} \left\| \left(T_{\mathrm{X}} W'^{-\top}\right)^{-1} \varepsilon\left(T_{\mathrm{X}} \left(W'^{-\top} y\right)\right) \right\|^2 \quad (14)
$$
$$
= \mathbb{E}_{y,\varepsilon(Ty)} \left\| T^{-1} \varepsilon\left(Ty\right) \right\|^2
$$

Therefore, the two-sided quantization problem in Eq. (10) is reduced to finding the optimal $T \in \mathbb{R}^{d \times d}$ that minimizes the one-sided quantization loss $\mathbb{E}_{y,\varepsilon(Ty)} \left\| T^{-1} \varepsilon\left(Ty\right) \right\|^2$ and compute

$$
T_{\mathrm{X}} = T W'^{\top}. \quad (15)
$$

**Transformation parameterization.** Let the unknown orthogonal matrices $U', R \in \mathbb{R}^{d \times d}$ and the unknown diagonal matrix $S' \in \mathbb{R}^{d \times d}$ be the singular value decomposition (SVD) of $TUS = U'S'R^\top$ with $U, S$ defined in Eq. (6); thus, $T$ can be parameterized as

$$
T = U'S'R^\top S^{-1} U^\top. \quad (16)
$$

Denote $S = \mathrm{diag}\left(s_1, \ldots, s_d\right)$ and $S' = \mathrm{diag}\left(s'_1, \ldots, s'_d\right)$. Without loss of generality, assume $s_1 \geq \cdots \geq s_d > 0$ and $s'_1 \geq \cdots \geq s'_d > 0$. A useful property of $S$ is

$$
\mathrm{tr}\left(S^2\right) \geq d^{-1}\left(\mathrm{tr}\left(S\right)\right)^2 \geq d^{-1}\mathrm{tr}\left(S^2\right) \quad (17)
$$

with detailed steps in Eq. (39). The equality $\mathrm{tr}\left(S^2\right) = d^{-1}\left(\mathrm{tr}\left(S\right)\right)^2$ is attained when all $s_i$ are the same (all singular values are inliers). The equality $d^{-1}\left(\mathrm{tr}\left(S\right)\right)^2 = d^{-1}\mathrm{tr}\left(S^2\right)$ is attained when $\mathrm{tr}\left(S\right) \to s_1$ (a singular value is an extreme outlier).

**Theorem 4.1** (Optimal Transform). *The optimal configuration of $T$ for floating-point (FP) data types is $U' = H$, $S' = S^{\frac{1}{2}}$, and $R = I$. For integer (INT) types, the same configuration is optimal up to a $d^{o(1)}$ factor for zero-mean multivariate Gaussian/Laplacian distributed data and within a $d$ factor for any distribution.*

*Proof.* We provide detailed proofs for FP and INT in Sections 4.2 and 4.3, respectively. For each data type, we apply a two-step proof strategy. We first (Sections 4.2.1 and 4.3.1) provide a smooth modeling of the quantizer and treat the type casting errors as random noise. Secondly (Sections 4.2.2 and 4.3.2), we calculate the expected quantization error, Eq. (14), with respect to the parameterized transform, Eq. (16), and minimize this expectation by solving the corresponding optimization problem. □

The optimal $T_{\mathrm{X}}$, namely the WUSH construction $T_{\mathrm{wush}}$ in Eq. (7), is obtained by applying Theorem 4.1 and Eqs. (15), (16).

## 4.2. Proof for Floating-Point (FP) Types

### 4.2.1. QUANTIZATION ERROR MODELING

For a number in FP types, the quantization error tends to be proportional to its absolute value. Formally, for a vector $\boldsymbol{\alpha} \in \mathbb{R}^d$, we can model the quantization error as

$$\varepsilon(\boldsymbol{\alpha}) = \mathrm{diag}(\boldsymbol{\eta})\boldsymbol{\alpha} \tag{18}$$

where $\boldsymbol{\eta} = [\eta_1, \ldots, \eta_d]^\top \in \mathbb{R}^d$ is a random vector of i.i.d. samples from the distribution $\eta \sim \mathcal{D}_\eta$ with $\mathbb{E}_\eta \eta = 0$. We provide more rigorous justifications for why this modeling makes sense in Appendix Section A.5.2.

### 4.2.2. LOSS MINIMIZATION

Using the quantization error modeling in Eq. (18), the minimization objective in Eq. (14) becomes

$$\begin{aligned}
&\mathbb{E}_{\boldsymbol{y}, \varepsilon(\boldsymbol{Ty})} \left\| \boldsymbol{T}^{-1} \varepsilon(\boldsymbol{Ty}) \right\|^2 \\
&= \left(\mathbb{E}_\eta \eta^2\right) \mathrm{tr}\left(\boldsymbol{T}^{-1}\left(\left(\boldsymbol{U}'\boldsymbol{S}'^2\boldsymbol{U}'^\top\right) \odot \boldsymbol{I}\right)\boldsymbol{T}^{-\top}\right)
\end{aligned} \tag{19}$$

with $\odot$ representing elementwise multiplication, and with detailed steps in Eq. (40). The lower bound of the trace term in Eq. (19) is

$$\mathrm{tr}\left(\boldsymbol{T}^{-1}\left(\left(\boldsymbol{U}'\boldsymbol{S}'^2\boldsymbol{U}'^\top\right) \odot \boldsymbol{I}\right)\boldsymbol{T}^{-\top}\right) \geq d^{-1}\left(\mathrm{tr}(\boldsymbol{S})\right)^2 \tag{20}$$

and the equality can be attained by choosing $\boldsymbol{U}' = \boldsymbol{H}$, $\boldsymbol{S}' = \boldsymbol{S}^{\frac{1}{2}}$, and $\boldsymbol{R} = \boldsymbol{I}$, with detailed steps in Eqs. (41), (42).

### 4.2.3. DISCUSSION

Under our smooth modeling in Eq. (18), for any orthogonal $\boldsymbol{T}$ (including the identity $\boldsymbol{T} = \boldsymbol{I}$ and the Hadamard $\boldsymbol{T} = \boldsymbol{H}$), the minimization objective becomes trivially

$$\mathbb{E}_{\boldsymbol{y}, \varepsilon(\boldsymbol{Ty})} \left\| \boldsymbol{T}^{-1}\varepsilon(\boldsymbol{Ty}) \right\|^2 = \left(\mathbb{E}_\eta \eta^2\right)\mathrm{tr}(\boldsymbol{S}^2) \tag{21}$$

with detailed steps in Eq. (43). Thus, orthogonal transforms will not be helpful for reducing the quantization error. Comparing the trace terms in Eqs. (20), (21) using the inequality Eq. (17), the WUSH transform can reduce the error by at most $d$ times in the extreme outlier scenario. Also note that the non-trivial choices $\boldsymbol{U}' = \boldsymbol{H}$ and $\boldsymbol{S}' = \boldsymbol{S}^{\frac{1}{2}}$ are both essential for reducing the error. Either trivial choices of $\boldsymbol{U}' = \boldsymbol{I}$ or $\boldsymbol{S}' = \boldsymbol{I}$ will lead to the same suboptimal trace term

$$\mathrm{tr}\left(\boldsymbol{T}^{-1}\left(\left(\boldsymbol{U}'\boldsymbol{S}'^2\boldsymbol{U}'^\top\right) \odot \boldsymbol{I}\right)\boldsymbol{T}^{-\top}\right) = \mathrm{tr}(\boldsymbol{S}^2) \tag{22}$$

as that in the case of the orthogonal $\boldsymbol{T}$ in Eq. (21), with detailed steps in Eqs. (44), (45). Figure 1 (left two columns) also visualizes these results.

## 4.3. Proof for Integer (INT) Data Types

### 4.3.1. QUANTIZATION ERROR MODELING

For a number in INT types, the quantization error tends to be proportional to the maximum absolute value within a quantization group. Formally, for a vector $\boldsymbol{\alpha} \in \mathbb{R}^d$, we can model the quantization error as

$$\varepsilon(\boldsymbol{\alpha}) = \|\boldsymbol{\alpha}\|_\infty \boldsymbol{\eta} \tag{23}$$

where $\boldsymbol{\eta} = [\eta_1, \ldots, \eta_d]^\top \in \mathbb{R}^d$ is a random vector of i.i.d. samples from the distribution $\eta \sim \mathcal{D}_\eta$ with $\mathbb{E}_\eta \eta = 0$. We provide more rigorous justifications for why this modeling makes sense in Appendix Section A.5.1.

### 4.3.2. LOSS MINIMIZATION

Using the quantization error modeling in Eq. (23), the minimization objective in Eq. (14) becomes

$$\mathbb{E}_{\boldsymbol{y}, \varepsilon(\boldsymbol{Ty})} \left\| \boldsymbol{T}^{-1}\varepsilon(\boldsymbol{Ty}) \right\|^2 = \left(\mathbb{E}_\eta \eta^2\right)\left\| \boldsymbol{T}^{-1} \right\|_{\mathrm{F}}^2 \mathbb{E}_{\boldsymbol{y}} \left\| \boldsymbol{Ty} \right\|_\infty^2 \tag{24}$$

with detailed steps in Eq. (46).

We can obtain a lower bound for the term $\left\| \boldsymbol{T}^{-1} \right\|_{\mathrm{F}}^2$ in Eq. (24).

$$\left\| \boldsymbol{T}^{-1} \right\|_{\mathrm{F}}^2 \geq \mathrm{tr}\left(\boldsymbol{S}^2\boldsymbol{S}'^{-2}\right), \tag{25}$$

and the equality is attained when $\boldsymbol{R} = \boldsymbol{I}$, with detailed steps in Eqs. (47), (48).

Next, we obtain near-optimal lower and upper bounds for the term $\mathbb{E}_{\boldsymbol{y}} \left\| \boldsymbol{Ty} \right\|_\infty^2$ in Eq. (24) by choosing $\boldsymbol{U}' = \boldsymbol{H}$.

$$\begin{aligned}
\mathbb{E}_{\boldsymbol{y}} \left\| \boldsymbol{Ty} \right\|_\infty^2 &\geq d^{-1}\mathrm{tr}\left(\boldsymbol{S}'^2\right), \\
\mathbb{E}_{\boldsymbol{y}} \left\| \boldsymbol{Ty} \right\|_\infty^2 &\leq \begin{cases} d^{o(1)-1}\mathrm{tr}\left(\boldsymbol{S}'^2\right), & \text{tail-bounded } \mathcal{D}_{\mathrm{y}}, \\ \mathrm{tr}\left(\boldsymbol{S}'^2\right), & \text{otherwise.} \end{cases}
\end{aligned} \tag{26}$$

Note that the upper bound is tighter when $\mathcal{D}_{\mathrm{y}}$ is a tail-bounded (zero-mean multivariate Gaussian or Laplacian) distribution than for a general distribution. For the full derivations of these bounds, please refer to Eqs. (49) to (58).

Taken Eqs. (25), (26) together, by setting $\boldsymbol{R} = \boldsymbol{I}$ and $\boldsymbol{U}' = \boldsymbol{H}$, the term $\left\| \boldsymbol{T}^{-1} \right\|_{\mathrm{F}}^2 \mathbb{E}_{\boldsymbol{y}} \left\| \boldsymbol{Ty} \right\|_\infty^2$ is bounded by $\mathrm{tr}\left(\boldsymbol{S}^2\boldsymbol{S}'^{-2}\right)\mathrm{tr}\left(\boldsymbol{S}'^2\right)$ within a gap of $d$ such that

$$\begin{aligned}
d^{-1}\mathrm{tr}\left(\boldsymbol{S}^2\boldsymbol{S}'^{-2}\right)\mathrm{tr}\left(\boldsymbol{S}'^2\right) &\leq \left\| \boldsymbol{T}^{-1} \right\|_{\mathrm{F}}^2 \mathbb{E}_{\boldsymbol{y}} \left\| \boldsymbol{Ty} \right\|_\infty^2 \\
&\leq \mathrm{tr}\left(\boldsymbol{S}^2\boldsymbol{S}'^{-2}\right)\mathrm{tr}\left(\boldsymbol{S}'^2\right).
\end{aligned} \tag{27}$$

And for a tail-bounded $\mathcal{D}_{\mathrm{y}}$, the gap is narrowed to $d^{o(1)}$ such that

$$\left\| \boldsymbol{T}^{-1} \right\|_{\mathrm{F}}^2 \mathbb{E}_{\boldsymbol{y}} \left\| \boldsymbol{Ty} \right\|_\infty^2 = d^{o(1)-1}\mathrm{tr}\left(\boldsymbol{S}^2\boldsymbol{S}'^{-2}\right)\mathrm{tr}\left(\boldsymbol{S}'^2\right). \tag{28}$$

By the Cauchy-Schwarz inequality, the lower bound of the trace product term is

$$\operatorname{tr}\big(\boldsymbol{S}^2 \boldsymbol{S}'^{-2}\big)\operatorname{tr}\big(\boldsymbol{S}'^2\big) \geq \Big(\operatorname{tr}\big(\big(\boldsymbol{S}^2\boldsymbol{S}'^{-2}\big)^{\frac{1}{2}}\big(\boldsymbol{S}'^2\big)^{\frac{1}{2}}\big)\Big)^2 \quad (29)$$
$$= \big(\operatorname{tr}\left(\boldsymbol{S}\right)\big)^2.$$

and the equality is attained when $\boldsymbol{S}' \propto \boldsymbol{S}^{\frac{1}{2}}$. Without loss of generality, we can choose $\boldsymbol{S}' = \boldsymbol{S}^{\frac{1}{2}}$ to minimize both the lower and upper bounds in Eqs. (27), (28), and the quantization loss in Eq. (24) becomes near-optimal:

$$d^{-1}\left(\operatorname{tr}\left(\boldsymbol{S}\right)\right)^2 \leq \big\|\boldsymbol{T}^{-1}\big\|_{\mathrm{F}}^2 \, \mathbb{E}_{\boldsymbol{y}} \, \|\boldsymbol{T}\boldsymbol{y}\|_\infty^2$$
$$\leq \begin{cases} d^{o(1)-1}\left(\operatorname{tr}\left(\boldsymbol{S}\right)\right)^2, & \text{tail-bounded } \mathcal{D}_{\mathrm{y}}, \\ \left(\operatorname{tr}\left(\boldsymbol{S}\right)\right)^2, & \text{otherwise.} \end{cases} \quad (30)$$

### 4.3.3. DISCUSSION

Consider the case of $\boldsymbol{T}$ being orthogonal. For any general distribution $\mathcal{D}_{\mathrm{y}}$, the bounds are

$$\operatorname{tr}\left(\boldsymbol{S}^2\right) \leq \big\|\boldsymbol{T}^{-1}\big\|_{\mathrm{F}}^2 \, \mathbb{E}_{\boldsymbol{y}} \, \|\boldsymbol{T}\boldsymbol{y}\|_\infty^2 \leq d\operatorname{tr}\left(\boldsymbol{S}^2\right) \quad (31)$$

with detailed steps in Eq. (59). For a tail-bounded $\mathcal{D}_{\mathrm{y}}$, the Hadamard $\boldsymbol{T} = \boldsymbol{H}$ is empirically the best orthogonal transform, so the bounds may be tightened to

$$\big\|\boldsymbol{T}^{-1}\big\|_{\mathrm{F}}^2 \, \mathbb{E}_{\boldsymbol{y}} \, \|\boldsymbol{T}\boldsymbol{y}\|_\infty^2 = d^{o(1)} \operatorname{tr}\left(\boldsymbol{S}^2\right) \quad (32)$$

Comparing Eq. (30) with Eqs. (31), (32) using the inequality Eq. (17), an orthogonal $\boldsymbol{T}$ is suboptimal, and the error bounds of the Hadamard $\boldsymbol{T} = \boldsymbol{H}$ can be at most $d$ times larger than those of the optimal $\boldsymbol{T}$, in the extreme outlier scenario. Figure 1 (right two columns) provides visualizations of different transforms.

## 5. GPU Kernel Support

As noted in Eq. (9), part of the WUSH transform has to be applied online during inference, requiring a specialized GPU kernel. Online blockwise *Hadamard* rotations are efficiently supported in MR-GPTQ's kernels (Egiazarian et al., 2026). Yet, in their case, the transforms are data-independent, so it is possible to reuse a single transform across all blocks. By contrast, WUSH assigns a *distinct* transform to each block. This per-block specialization significantly complicates kernel reuse and fusion in existing low-level libraries.

**Fused WUSH + Quant kernel implementation.** A first design decision is the storage layout of the WUSH matrices. For block size G and number of blocks C, we store matrices as `(G,G,C)`, i.e., transposed with respect to the channel dimension. Since this transposition is performed offline, it does not introduce any runtime overhead. This layout

choice is motivated by the observation that, for small group sizes (e.g., `G=32` for MXFP4), the WUSH transformation is memory-bound. Accordingly, the proposed kernel formulates the operation as a GEMM-equivalent computation, corresponding to C independent dense transforms. Each thread block processes a `Tile_M×G` subproblem, where `Tile_M` denotes the tile size along the batch dimension. Crucially, each block only needs to load a single `(G,G)` matrix to produce its output, matching the Hadamard case (H + Quant) in MR-GPTQ, maximizing effective bandwidth utilization.

The kernel is implemented using a CUTLASS GEMM template. The outer dimension M of the activation tensor (with shape `(M,K)`) and the outer dimension G of the WUSH matrix (with shape `(G,G,C)`) are mapped to the M and N dimensions of the GEMM, respectively. The internal GEMM dimension K' is fixed to G, rather than G×C. The index over C is instead handled implicitly as an offset that each thread block applies based on the specific subproblem it is assigned to.

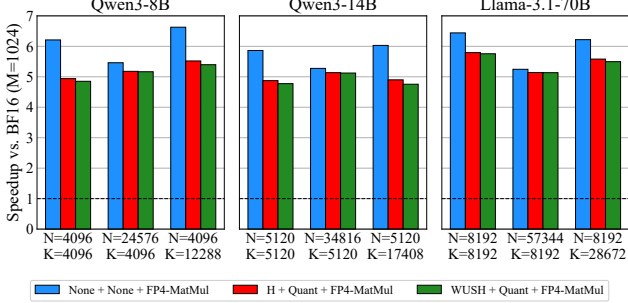

*Figure 2.* **Per-layer MXFP4 (group size `G=32`) inference speedups relative to BF16 at batch size M=1024 for Qwen3-8B, Qwen3-14B, and Llama-3.1-70B.** We compare FP4-MatMul kernel without transform or quantization (None + None), with fused Hadamard and quantization kernel (H + Quant), and with fused WUSH and quantization kernel (WUSH + Quant). For block size G and a layer with K=C×G input and N output channels, the Hadamard matrix size is `(G,G)`, while the WUSH size is `(G,G,C)`. For H + Quant, the QuTLASS implementation is used (Egiazarian et al., 2026), which also fuses the transform with the quantization step.

**Performance.** Figure 2 reports per-layer inference speedups on RTX 5090 at batch size 1024 for Qwen3-8B, Qwen3-14B, and Llama-3.1-70B, normalized to BF16. Across all models and layer shapes, the MXFP4 matrix multiplication kernel (FP4-MatMul) achieves speedups of up to 6.6× over BF16, compared to the theoretical peak of 8× on this GPU. When including the fused transform and quantization kernel, WUSH + Quant + FP4-MatMul achieves speedups of up to 5.8× over BF16, and it closely matches the performance of the optimized H + Quant + FP4-MatMul baseline, with an average throughput difference of only 1.3%. Remarkably, in some configurations, WUSH + Quant + FP4-MatMul achieves identical performance to H + Quant + FP4-MatMul.

This negligible difference arises from loading C transform matrices instead of a single one; however, it is almost entirely mitigated by the kernel design and effective reuse via the L1 and L2 caches. Pure-quantization without an online transform (None + Quant + FP4-MatMul) would remove the transform cost but still incur the same activation quantization overhead, and therefore its performance is expected to lie between None + None + FP4-MatMul and H + Quant + FP4-MatMul. These results demonstrate that WUSH introduces negligible additional costs.

## 6. Experiments

**Setup.** We now present experiments evaluating the layerwise losses and the accuracy of quantized models using the WUSH transform, relative to SmoothQuant (Xiao et al., 2023), QuaRot (Ashkboos et al., 2024b), SpinQuant (Liu et al., 2025), and MR-GPTQ (Egiazarian et al., 2026) baselines. MR-GPTQ is the current state-of-the-art for MXFP/NVFP quantization. Beyond comparisons to prior methods, these experiments also include ablations over both the transform design and the quantization procedure. Specifically, we compare progressively richer transform variants to isolate the effects of orthogonal mixing, the Hadamard backbone, and the adaptive non-orthogonal component, while also evaluating both RTN and GPTQ quantization. We conduct experiments on the Llama-3.2-3B-Instruct, Llama-3.1-8B-Instruct, and Qwen3-8B/14B/32B models. We use Platinum Benchmarks (Vendrow et al., 2025) and the LM Evaluation Harness (Gao et al., 2021) for accuracy evaluation.

**Offline preprocessing costs.** We report empirical offline preprocessing costs (Table 6) and storage overheads (Table 7) in Appendix Section B.2.

### 6.1. Superior Layerwise Quantization Loss

We first report the layerwise weight-activation RTN quantization loss ($L_2$ loss normalized by the number of elements) for each linear layer (attention projection layers Q, K, V, O and MLP projection layers Gate (G), Up (U), Down (D)) in the 18th transformer block of Qwen3-8B, using 32 calibration samples of sequence length 2048 from the FineWeb-Edu dataset (Penedo et al., 2024). We found the results to be consistent across blocks and datasets. We compare the WUSH transform with the identity (I), random rotation (R) averaged over 10 runs, Hadamard (H), and WUSH without Hadamard (WUS). Table 1 summarizes the losses for MXFP4, NVFP4, and INT4 formats. Following the format definitions, MXFP4 uses group size 32, and NVFP4 uses group size 16. We use INT4 to denote 4-bit integers with Gaussian MSE clipping and BF16 scales of group size 32. The transform block size is matched to the quantization group size. Across formats, WUSH substantially reduces

*Table 1.* Layerwise RTN quantization loss in the unit of $10^{-3}$.

| | | Q | K | V | O | G | U | D |
|---|---|---|---|---|---|---|---|---|
| **MXFP4** | I | 11.1 | 12.0 | 10.7 | 4.35 | 7.10 | 6.56 | 5.47 |
| | R | 7.61 | 7.73 | 9.14 | 3.84 | 5.56 | 5.68 | 4.07 |
| | H | 7.24 | 7.20 | 8.60 | 3.79 | 5.45 | 5.61 | 3.90 |
| | WUS | 6.27 | 7.22 | 4.05 | 3.57 | 5.76 | 4.75 | 4.46 |
| | WUSH | **3.34** | **3.34** | **3.30** | **2.76** | **4.49** | **4.39** | **3.39** |
| **NVFP4** | I | 4.23 | 4.35 | 4.37 | 2.34 | 3.49 | 3.41 | 2.41 |
| | R | 4.98 | 5.01 | 5.86 | 2.54 | 3.63 | 3.68 | 2.66 |
| | H | 5.60 | 5.62 | 6.70 | 2.58 | 3.71 | 3.79 | 2.78 |
| | WUS | **2.26** | **2.36** | 2.30 | 2.00 | **3.04** | **3.01** | **2.23** |
| | WUSH | 2.40 | 2.44 | **2.28** | **1.92** | 3.09 | 3.02 | 2.33 |
| **INT4** | I | 170. | 123. | 13.2 | 4.55 | 55.9 | 9.83 | 19.3 |
| | R | 5.79 | 5.84 | 6.96 | 2.94 | 4.22 | 4.29 | 3.10 |
| | H | 5.57 | 5.55 | 6.80 | 2.86 | 4.09 | 4.25 | 3.03 |
| | WUS | 213. | 142. | 10.7 | 4.54 | 50.2 | 7.42 | 13.1 |
| | WUSH | **2.39** | **2.43** | **2.54** | **2.10** | **3.43** | **3.43** | **2.55** |

the loss: WUSH consistently yields the smallest loss for MXFP4 and INT4, while WUSH and WUS are almost equal for NVFP4. The errors for NVFP4 tend to be lower due to its smaller group size.

Although the theoretical derivation uses a stochastic unbiased quantization model as a tractable surrogate, the resulting transform is applied with deterministic RTN in these layerwise experiments. The trends in Table 1 support this surrogate, suggesting that the assumptions (Section 4.1) and modeling (Sections 4.2.1 and 4.3.1) capture the dominant sources of error for FP and INT block quantizers. At the same time, MXFP4 behaves as a hybrid between ideal FP and INT quantization: the effective mantissa step changes uniformly, imparting INT-like behavior, especially in the subnormal regime and small exponent ranges. This explains why Hadamard transforms provide measurable gains for MXFP4, despite the ideal FP theory (Section 4.2.3) predicting no benefit for purely orthogonal transforms. The poor INT4 performance of WUS further highlights the role of the Hadamard component: without it, the non-orthogonal component can amplify individual coordinates, increasing the AbsMax group scale and therefore the INT quantization error. For NVFP4, the Hadamard alone is harmful due to the top-element preservation effect of this format (Egiazarian et al., 2026). Remarkably, other components in WUSH overcome this effect and produce smaller losses than the identity transform.

### 6.2. End-to-End LLM Accuracy Benchmarks

Table 2 compares WUSH relative to prior works for Llama-3.1-8B-Instruct on LM Evaluation Harness metrics, in the same setup as Egiazarian et al. (2026). For methods with block transforms (i.e., not applicable to SmoothQuant, QuaRot, and SpinQuant), the transform block size is al-

*Table 2.* Llama-3.1-8B-Instruct W4A4 accuracy results on the LM Evaluation Harness under different methods.

| Format | Method | MMLU-CoT | GSM8K | HellaSwag | WinoGrande | Average | Recovery |
|--------|--------|----------|-------|-----------|------------|---------|----------|
| BF16 | - | 72.76 | 85.06 | 80.01 | 77.90 | 78.93 | 100.0 |
| NVFP4 | RTN-I | 68.26 | 78.39 | 78.15 | 74.11 | 74.73 | 94.67 |
| | RTN-H | 67.41 | 78.01 | 77.31 | 73.48 | 74.05 | 93.82 |
| | SmoothQuant | 68.90 | 79.50 | 79.50 | 74.70 | 75.70 | 95.90 |
| | QuaRot | 66.50 | 77.40 | 77.25 | 75.14 | 74.10 | 93.80 |
| | SpinQuant | 66.50 | 76.10 | 76.96 | 75.32 | 73.70 | 93.40 |
| | GPTQ-I | 68.85 | 81.25 | 78.26 | 74.51 | 75.72 | 95.92 |
| | GPTQ-H (MR-GPTQ) | 69.12 | 80.80 | 78.17 | 75.24 | 75.84 | 96.08 |
| | RTN-WUSH | 68.83 | 78.57 | 78.22 | 75.47 | 75.28 | 95.37 |
| | GPTQ-WUSH | 69.69 | 80.11 | 78.52 | 76.09 | **76.10** | **96.40** |
| MXFP4 | RTN-I | 62.21 | 67.85 | 73.99 | 73.24 | 69.32 | 87.83 |
| | RTN-H | 62.38 | 72.48 | 75.29 | 71.67 | 70.45 | 89.26 |
| | SmoothQuant | 63.93 | 68.54 | 75.10 | 73.56 | 70.30 | 89.06 |
| | QuaRot | 49.86 | 56.94 | 73.50 | 71.43 | 62.90 | 79.70 |
| | SpinQuant | 61.80 | 68.16 | 74.87 | 72.93 | 69.40 | 88.00 |
| | GPTQ-I | 63.49 | 68.46 | 76.01 | 74.51 | 70.62 | 89.47 |
| | GPTQ-H (MR-GPTQ) | 67.19 | 75.70 | 76.91 | 74.80 | 73.65 | 93.31 |
| | RTN-WUSH | 66.85 | 75.16 | 77.28 | 73.56 | 73.21 | 92.75 |
| | GPTQ-WUSH | 67.79 | 77.41 | 77.44 | 74.78 | **74.35** | **94.20** |

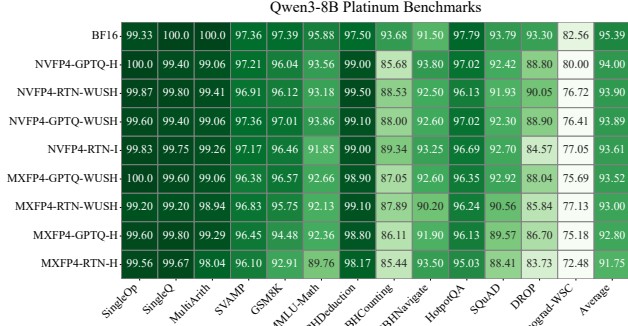

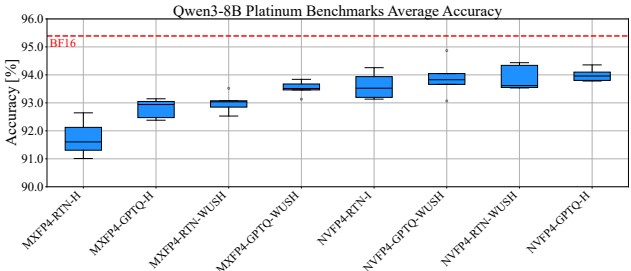

*Figure 3.* Comparison of different transforms on Qwen3-8B for both NVFP4 and MXFP4 quantization on Platinum Benchmarks. The top table shows accuracy results across the individual benchmark tasks, while the bottom plot shows the average accuracy scores together with their standard deviations for each transform.

ways the same as the quantization group size. Overall, WUSH generally yields the best results, by a large margin for MXFP4. On Qwen3-8B measured on Platinum Benchmarks (Figure 3), WUSH improves MXFP4 by up to +1.3 (RTN) and +0.7 (GPTQ) average points, while NVFP4 improvements are smaller and often within run-to-run variability. Additional accuracy results are in Appendix Section B.1. Table 3 shows the LM Evaluation Harness results for different models, e.g., for Qwen3-14B WUSH reduces the NVFP4-MXFP4 average gap to 0.36 points, and above 98% recovery. Figures 7 to 10 also reflect similar trends. Table 4 shows WUSH reduces the KL divergence. Table 5 shows WUSH is stable with respect to the choice of calibration dataset. We conclude that WUSH improves the state-of-the-art for weight-activation quantization.

## 7. Conclusion

We have studied weight-activation quantization for LLMs in the presence of heavy-tailed outliers and have shown that this problem admits closed-form linear transforms for both FP and INT block quantizers. Our construction, WUSH, combines a fixed Hadamard backbone with a data-aware component derived from second-order statistics, yielding a non-orthogonal blockwise transform that is provably near-optimal and remains amenable to efficient GPU kernel implementations. These results clarify the empirical success of Hadamard rotations and indicate that principled, data-aware block transforms can yield significant improvements, e.g., making the MXFP format competitive with NVFP.

## Acknowledgements

We thank Ileana Rugina for useful discussions. We thank Tijmen Blankevoort for useful discussions and for suggesting the name WUSH, which is a clear improvement over our previous name (HSUW). This research was funded in part by the Austrian Science Fund (FWF) 10.55776/COE12, and partially supported by a generous grant from NVIDIA.

## Impact Statement

This paper presents work whose goal is to advance the field of machine learning. There are many potential societal consequences of our work, none of which we feel must be specifically highlighted here.

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

# A. Detailed Theoretical Results

## A.1. Second Moment Decompositions

**Lemma A.1** (Invariance of the Decomposition Choices). *For any full rank $\boldsymbol{W}'_{(i)}, \boldsymbol{X}'_{(i)}$ that satisfy Eq. (5), the matrix products $\boldsymbol{W}'_{(i)}\boldsymbol{U}_{(i)}$ and $\boldsymbol{X}'_{(i)}\boldsymbol{V}_{(i)}$ in Eq. (7) are invariant up to sign and permutation changes.*

*Proof.* We omit the subscript $(i)$ and superscript $'$ to simplify notation.

Let $\widehat{\boldsymbol{W}}, \widehat{\boldsymbol{X}} \in \mathbb{R}^{d \times d}$ such that $\widehat{\boldsymbol{W}}\widehat{\boldsymbol{W}}^\top = \boldsymbol{W}\boldsymbol{W}^\top$ and $\widehat{\boldsymbol{X}}\widehat{\boldsymbol{X}}^\top = \boldsymbol{X}\boldsymbol{X}^\top$. Define $\boldsymbol{Q}_\mathrm{W} = \boldsymbol{W}^{-1}\widehat{\boldsymbol{W}}$ and $\boldsymbol{Q}_\mathrm{X} = \boldsymbol{X}^{-1}\widehat{\boldsymbol{X}}$. Then,

$$\boldsymbol{Q}_\mathrm{W}\boldsymbol{Q}_\mathrm{W}^\top = \boldsymbol{W}^{-1}\widehat{\boldsymbol{W}}\widehat{\boldsymbol{W}}^\top\boldsymbol{W}^{-\top} = \boldsymbol{W}^{-1}(\boldsymbol{W}\boldsymbol{W}^\top)\boldsymbol{W}^{-\top} = \mathbf{I}, \tag{33}$$

and similarly $\boldsymbol{Q}_\mathrm{X}\boldsymbol{Q}_\mathrm{X}^\top = \mathbf{I}$. Hence $\boldsymbol{Q}_\mathrm{W}, \boldsymbol{Q}_\mathrm{X}$ are orthogonal and $\widehat{\boldsymbol{W}} = \boldsymbol{W}\boldsymbol{Q}_\mathrm{W}, \widehat{\boldsymbol{X}} = \boldsymbol{X}\boldsymbol{Q}_\mathrm{X}$. By Eq. (6), we have $\boldsymbol{U}, \boldsymbol{S}, \boldsymbol{V} \in \mathbb{R}^{d \times d}$ as the SVD of $\boldsymbol{W}^\top\boldsymbol{X} = \boldsymbol{U}\boldsymbol{S}\boldsymbol{V}^\top$. Therefore,

$$\widehat{\boldsymbol{W}}^\top\widehat{\boldsymbol{X}} = \boldsymbol{Q}_\mathrm{W}^\top\boldsymbol{W}^\top\boldsymbol{X}\boldsymbol{Q}_\mathrm{X} = \boldsymbol{Q}_\mathrm{W}^\top\left(\boldsymbol{U}\boldsymbol{S}\boldsymbol{V}^\top\right)\boldsymbol{Q}_\mathrm{X} = \left(\boldsymbol{Q}_\mathrm{W}^\top\boldsymbol{U}\right)\boldsymbol{S}\left(\boldsymbol{Q}_\mathrm{X}^\top\boldsymbol{V}\right)^\top. \tag{34}$$

Thus, we may take $\widehat{\boldsymbol{U}} = \boldsymbol{Q}_\mathrm{W}^\top\boldsymbol{U}, \widehat{\boldsymbol{S}} = \boldsymbol{S}, \widehat{\boldsymbol{V}} = \boldsymbol{Q}_\mathrm{X}^\top\boldsymbol{V}$ (up to the sign and permutation conventions) as the SVD of $\widehat{\boldsymbol{W}}^\top\widehat{\boldsymbol{X}} = \widehat{\boldsymbol{U}}\widehat{\boldsymbol{S}}\widehat{\boldsymbol{V}}^\top$. With this,

$$\begin{aligned}\widehat{\boldsymbol{W}}\widehat{\boldsymbol{U}} &= \boldsymbol{W}\boldsymbol{Q}_\mathrm{W}\boldsymbol{Q}_\mathrm{W}^\top\boldsymbol{U} = \boldsymbol{W}\boldsymbol{U}, \\ \widehat{\boldsymbol{X}}\widehat{\boldsymbol{V}} &= \boldsymbol{X}\boldsymbol{Q}_\mathrm{X}\boldsymbol{Q}_\mathrm{X}^\top\boldsymbol{V} = \boldsymbol{X}\boldsymbol{V}.\end{aligned} \tag{35}$$

$\square$

## A.2. Multivariate vs. Univariate Probabilistic Modeling

A common intuition in prior transform-based quantization methods is largely univariate: apply an orthogonal mixing (Hadamard or learned rotation) so that the distribution of coordinate-wise values becomes more "Gaussian-like," thereby reducing the impact of heavy-tailed outliers. In contrast, our probabilistic modeling is explicitly multivariate: we model a quantization group as a joint random vector and reason about its second-order structure, enabling analysis in the eigenbasis and construction of transforms that depend on the spectrum (eigenvalues / singular values).

Figure 4 illustrates the key limitation of purely orthogonal "Gaussianization." In the top row (original distributions), the joint density is anisotropic: the two coordinate-wise marginals differ in spread, and their average can appear heavy-tailed or non-Gaussian depending on the axis alignment. In the bottom row, after the Hadamard ($45°$ rotation) transform, the coordinate marginals become nearly identical, and the average marginal appears more Gaussian-like. However, this does not imply that outliers have been removed in the underlying multivariate geometry: orthogonal transforms preserve global energy and, crucially, preserve the eigenvalues of the covariance (they only rotate eigenvectors). Consequently, points that are extreme in the principal directions remain extreme in the joint space, even if they are less visually apparent in any single marginal.

This distinction is central to group quantization. Quantization error is driven by joint properties (e.g., how mass concentrates along dominant directions and how extreme points interact with groupwise scaling), which cannot be certified by marginal statistics alone. Our multivariate formulation therefore targets the spectrum directly: by exposing and using the eigen-/singular-value structure of the group statistics, we can design transforms that genuinely reduce the effective anisotropy responsible for outlier-dominated scaling, rather than merely redistributing it across coordinates via rotations.

## A.3. Equalization Effect

Figure 5 provides a layerwise visualization of the equalization effect induced by the WUSH construction. Using Eqs. (5), (7), for each block, the second moment of the activation after the WUSH transform satisfies

$$d_\mathrm{batch}^{-1}\boldsymbol{T}_{\mathrm{wush}(i)}\boldsymbol{X}_{(i)}\boldsymbol{X}_{(i)}^\top\boldsymbol{T}_{\mathrm{wush}(i)}^\top = \boldsymbol{T}_{\mathrm{wush}(i)}\boldsymbol{X}'_{(i)}\boldsymbol{X}'^\top_{(i)}\boldsymbol{T}_{\mathrm{wush}(i)}^\top = \boldsymbol{H}\boldsymbol{S}_{(i)}\boldsymbol{H}^\top. \tag{36}$$

Since the normalized Hadamard matrix has entries $\pm d^{-\frac{1}{2}}$, the diagonal of the transformed second moment is a constant of $d^{-1}\operatorname{tr}\left(\boldsymbol{S}_{(i)}\right)$. The analogous property also holds for the transformed weight second moment, so WUSH equalizes the

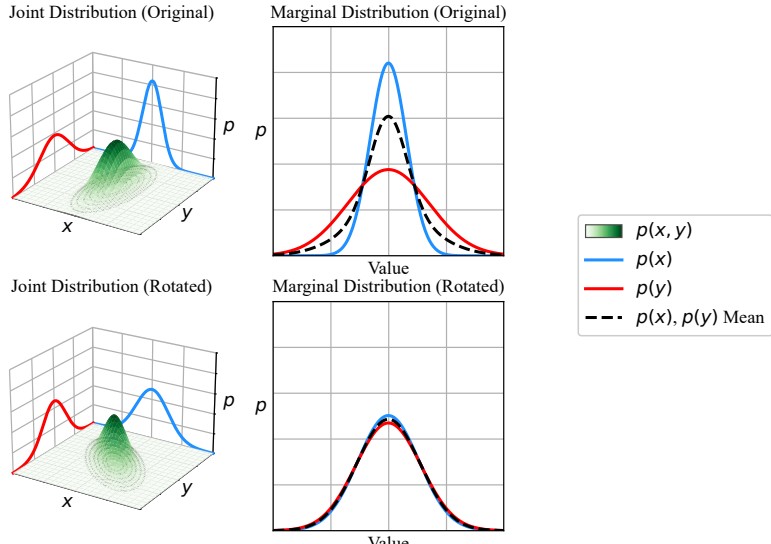

*Figure 4.* **Rotation can equalize marginals without alleviating multivariate outliers.** Top: an anisotropic 2D distribution with unequal coordinate marginals. Bottom: after an orthogonal rotation, the two marginals become nearly identical, and their average appears more Gaussian-like. Nevertheless, the joint distribution retains the same eigenvalues (spectrum) under rotation, so extreme points remain extreme in the multivariate space; they are merely "hidden" from any single marginal view. This motivates multivariate, spectrum-aware transforms that act on the joint geometry rather than relying on marginal "Gaussianization."

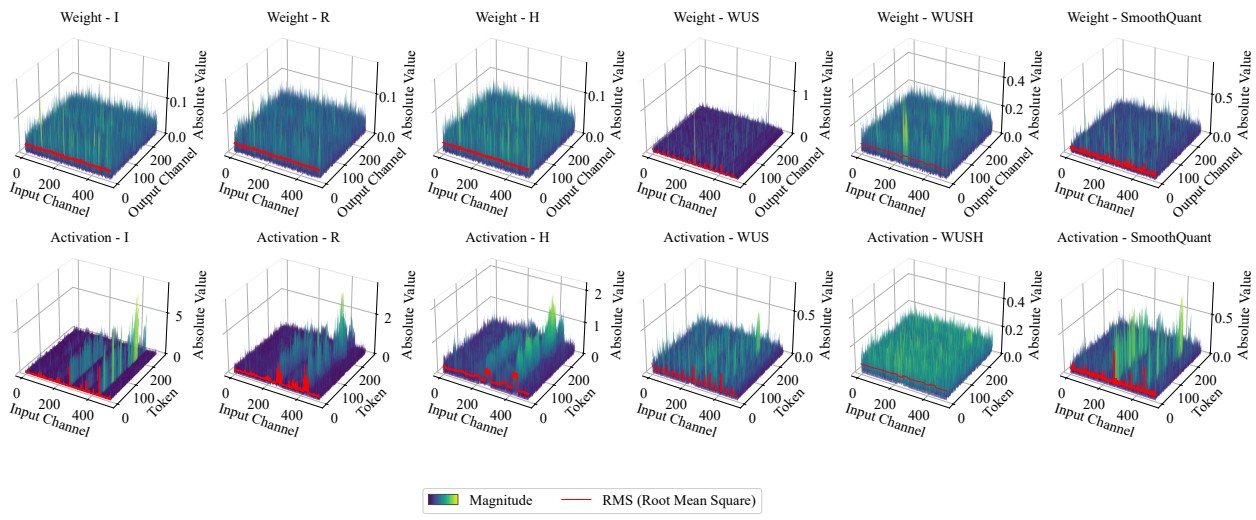

*Figure 5.* **Visualization of blockwise RMS equalization induced by WUSH.** Each panel uses its own vertical scale for visibility. We visualize absolute values of a representative weight slice (top row; input channel vs. output channel) and calibration activation slice (bottom row; input channel vs. token) after applying identity (I), random rotation (R), Hadamard (H), WUS (WUSH without Hadamard), WUSH, and SmoothQuant transforms. The surfaces show entrywise magnitudes, while the red curves show the per-input-channel RMS across output channels or tokens. Identity leaves the original outliers unchanged. Random rotation and Hadamard spread outliers within each weight or activation block, but as orthogonal transforms, they cannot transfer scale between weights and activations. Hadamard nevertheless equalizes RMS within a block more effectively than a random rotation. SmoothQuant transfers scale between activations and weights through diagonal channelwise rescaling, but can unevenly amplify some channels. WUS applies the adaptive non-orthogonal component but removes the Hadamard factor, leaving the transformed second moment aligned with $S$ and the per-channel RMS unequal. In contrast, WUSH maps each blockwise second moment to $HSH^{\top}$, whose diagonal is constant because $H$ is a normalized Hadamard matrix. Therefore, WUSH balances the weight and activation scales and produces the flattest RMS profiles.

per-channel RMS (root mean square) within the corresponding weight and activation blocks. This distinguishes WUSH from orthogonal transforms, which can spread outliers but cannot transfer scale between weights and activations, and from SmoothQuant, which transfers scale only through diagonal channelwise rescaling. Consistent with this, Figure 5 shows that WUSH balances the weight and activation scales and produces the flattest RMS profiles.

### A.4. Equation Derivations

**Standard basis vector notation.**

Denote the $k$-th standard basis vector as $\mathbf{e}_k \in \{0, 1\}^d$, where the $k$-th element is 1 and 0 otherwise.

**Basic second-order expectations.**

$$
\begin{aligned}
\mathbb{E}_{\boldsymbol{y}} \boldsymbol{y} \boldsymbol{y}^\top &= \boldsymbol{W}'^\top \left( \mathbb{E}_{\boldsymbol{x}} \boldsymbol{x} \boldsymbol{x}^\top \right) \boldsymbol{W}' = \boldsymbol{W}'^\top \boldsymbol{X}' \boldsymbol{X}'^\top \boldsymbol{W}' = \boldsymbol{U} \boldsymbol{S} \boldsymbol{V}^\top \boldsymbol{V} \boldsymbol{S} \boldsymbol{U}^\top = \boldsymbol{U} \boldsymbol{S}^2 \boldsymbol{U}^\top, && \text{(Eqs. (6), (11))} \\
\mathbb{E}_{\boldsymbol{y}} \boldsymbol{T} \boldsymbol{y} \left( \boldsymbol{T} \boldsymbol{y} \right)^\top &= \boldsymbol{T} \left( \mathbb{E}_{\boldsymbol{y}} \boldsymbol{y} \boldsymbol{y}^\top \right) \boldsymbol{T}^\top = \boldsymbol{U}' \boldsymbol{S}' \boldsymbol{R}^\top \boldsymbol{S}^{-1} \boldsymbol{U}^\top \boldsymbol{U} \boldsymbol{S}^2 \boldsymbol{U}^\top \boldsymbol{U} \boldsymbol{S}^{-1} \boldsymbol{R} \boldsymbol{S}' \boldsymbol{U}'^\top = \boldsymbol{U}' \boldsymbol{S}'^2 \boldsymbol{U}'^\top, && \text{(Eq. (16))} \\
\mathbb{E}_{\boldsymbol{y}} \| \boldsymbol{y} \|^2 &= \operatorname{tr} \left( \mathbb{E}_{\boldsymbol{y}} \boldsymbol{y} \boldsymbol{y}^\top \right) = \operatorname{tr} \left( \boldsymbol{U} \boldsymbol{S}^2 \boldsymbol{U}^\top \right) = \operatorname{tr} \left( \boldsymbol{S}^2 \right), \\
\mathbb{E}_{\boldsymbol{y}} \| \boldsymbol{T} \boldsymbol{y} \|^2 &= \operatorname{tr} \left( \boldsymbol{T} \boldsymbol{y} \left( \boldsymbol{T} \boldsymbol{y} \right)^\top \right) = \operatorname{tr} \left( \boldsymbol{U}' \boldsymbol{S}'^2 \boldsymbol{U}'^\top \right) = \operatorname{tr} \left( \boldsymbol{S}'^2 \right).
\end{aligned}
\tag{37}
$$

**Eq. (13).**

$$
\begin{aligned}
\ell &= \mathbb{E}_{\boldsymbol{w}, \boldsymbol{x}, \varepsilon(\boldsymbol{T}_\mathrm{W} \boldsymbol{w}), \varepsilon(\boldsymbol{T}_\mathrm{X} \boldsymbol{x})} \left( \left( \boldsymbol{T}_\mathrm{W} \boldsymbol{w} + \varepsilon \left( \boldsymbol{T}_\mathrm{W} \boldsymbol{w} \right) \right)^\top \left( \boldsymbol{T}_\mathrm{X} \boldsymbol{x} + \varepsilon \left( \boldsymbol{T}_\mathrm{X} \boldsymbol{x} \right) \right) - \boldsymbol{w}^\top \boldsymbol{x} \right)^2 \\
&= \mathbb{E}_{\boldsymbol{w}, \boldsymbol{x}, \varepsilon(\boldsymbol{T}_\mathrm{W} \boldsymbol{w}), \varepsilon(\boldsymbol{T}_\mathrm{X} \boldsymbol{x})} \Big( \boldsymbol{x}^\top \boldsymbol{T}_\mathrm{X}^\top \varepsilon \left( \boldsymbol{T}_\mathrm{W} \boldsymbol{w} \right) + \boldsymbol{w}^\top \boldsymbol{T}_\mathrm{W}^\top \varepsilon \left( \boldsymbol{T}_\mathrm{X} \boldsymbol{x} \right) + \underbrace{\varepsilon \left( \boldsymbol{T}_\mathrm{W} \boldsymbol{w} \right)^\top \varepsilon \left( \boldsymbol{T}_\mathrm{X} \boldsymbol{x} \right)}_{\approx\, 0 \quad \text{(first order approx.)}} + \underbrace{\boldsymbol{w}^\top \boldsymbol{T}_\mathrm{W}^\top \boldsymbol{T}_\mathrm{X} \boldsymbol{x} - \boldsymbol{w}^\top \boldsymbol{x}}_{=\, 0 \quad (\boldsymbol{T}_\mathrm{W} = \boldsymbol{T}_\mathrm{X}^{-\top})} \Big)^2 \\
&\approx \mathbb{E}_{\boldsymbol{w}, \boldsymbol{x}, \varepsilon(\boldsymbol{T}_\mathrm{W} \boldsymbol{w}), \varepsilon(\boldsymbol{T}_\mathrm{X} \boldsymbol{x})} \left( \boldsymbol{x}^\top \boldsymbol{T}_\mathrm{X}^\top \varepsilon \left( \boldsymbol{T}_\mathrm{W} \boldsymbol{w} \right) + \boldsymbol{w}^\top \boldsymbol{T}_\mathrm{W}^\top \varepsilon \left( \boldsymbol{T}_\mathrm{X} \boldsymbol{x} \right) \right)^2 \\
&= \mathbb{E}_{\boldsymbol{w}, \boldsymbol{x}, \varepsilon(\boldsymbol{T}_\mathrm{W} \boldsymbol{w}), \varepsilon(\boldsymbol{T}_\mathrm{X} \boldsymbol{x})} \left( \boldsymbol{x}^\top \boldsymbol{T}_\mathrm{X}^\top \varepsilon \left( \boldsymbol{T}_\mathrm{W} \boldsymbol{w} \right) \right)^2 + \left( \boldsymbol{w}^\top \boldsymbol{T}_\mathrm{W}^\top \varepsilon \left( \boldsymbol{T}_\mathrm{X} \boldsymbol{x} \right) \right)^2 + \left( \boldsymbol{x}^\top \boldsymbol{T}_\mathrm{X}^\top \underbrace{\varepsilon \left( \boldsymbol{T}_\mathrm{W} \boldsymbol{w} \right)}_{\mathbb{E}(\cdot)\,=\,0} \right) \left( \boldsymbol{w}^\top \boldsymbol{T}_\mathrm{W}^\top \underbrace{\varepsilon \left( \boldsymbol{T}_\mathrm{X} \boldsymbol{x} \right)}_{\mathbb{E}(\cdot)\,=\,0} \right) \\
&= \mathbb{E}_{\boldsymbol{w}, \boldsymbol{x}, \varepsilon(\boldsymbol{T}_\mathrm{W} \boldsymbol{w}), \varepsilon(\boldsymbol{T}_\mathrm{X} \boldsymbol{x})} \left( \boldsymbol{x}^\top \boldsymbol{T}_\mathrm{X}^\top \varepsilon \left( \boldsymbol{T}_\mathrm{W} \boldsymbol{w} \right) \right)^2 + \left( \boldsymbol{w}^\top \boldsymbol{T}_\mathrm{W}^\top \varepsilon \left( \boldsymbol{T}_\mathrm{X} \boldsymbol{x} \right) \right)^2 \\
&= \left( \mathbb{E}_{\boldsymbol{w}, \boldsymbol{x}, \varepsilon(\boldsymbol{T}_\mathrm{W} \boldsymbol{w})} \varepsilon \left( \boldsymbol{T}_\mathrm{W} \boldsymbol{w} \right)^\top \boldsymbol{T}_\mathrm{X} \underbrace{\boldsymbol{x} \boldsymbol{x}^\top}_{\mathbb{E}_{\boldsymbol{x}} \boldsymbol{x} \boldsymbol{x}^\top = \boldsymbol{X}' \boldsymbol{X}'^\top} \boldsymbol{T}_\mathrm{X}^\top \varepsilon \left( \boldsymbol{T}_\mathrm{W} \boldsymbol{w} \right) \right) + \left( \mathbb{E}_{\boldsymbol{w}, \boldsymbol{x}, \varepsilon(\boldsymbol{T}_\mathrm{X} \boldsymbol{x})} \varepsilon \left( \boldsymbol{T}_\mathrm{X} \boldsymbol{x} \right)^\top \boldsymbol{T}_\mathrm{W} \underbrace{\boldsymbol{w} \boldsymbol{w}^\top}_{\mathbb{E}_{\boldsymbol{w}} \boldsymbol{w} \boldsymbol{w}^\top = \boldsymbol{W}' \boldsymbol{W}'^\top} \boldsymbol{T}_\mathrm{W}^\top \varepsilon \left( \boldsymbol{T}_\mathrm{X} \boldsymbol{x} \right) \right) \\
&= \left( \mathbb{E}_{\boldsymbol{w}, \varepsilon(\boldsymbol{T}_\mathrm{W} \boldsymbol{w})} \varepsilon \left( \boldsymbol{T}_\mathrm{W} \boldsymbol{w} \right)^\top \boldsymbol{T}_\mathrm{X} \boldsymbol{X}' \boldsymbol{X}'^\top \boldsymbol{T}_\mathrm{X}^\top \varepsilon \left( \boldsymbol{T}_\mathrm{W} \boldsymbol{w} \right) \right) + \left( \mathbb{E}_{\boldsymbol{x}, \varepsilon(\boldsymbol{T}_\mathrm{X} \boldsymbol{x})} \varepsilon \left( \boldsymbol{T}_\mathrm{X} \boldsymbol{x} \right)^\top \boldsymbol{T}_\mathrm{W} \boldsymbol{W}' \boldsymbol{W}'^\top \boldsymbol{T}_\mathrm{W}^\top \varepsilon \left( \boldsymbol{T}_\mathrm{X} \boldsymbol{x} \right) \right) \\
&= \left( \mathbb{E}_{\boldsymbol{w}, \varepsilon(\boldsymbol{T}_\mathrm{W} \boldsymbol{w})} \big\| \boldsymbol{X}'^\top \underbrace{\boldsymbol{T}_\mathrm{X}^\top}_{=\, \boldsymbol{T}_\mathrm{W}^{-1}} \varepsilon \left( \boldsymbol{T}_\mathrm{W} \boldsymbol{w} \right) \big\|^2 \right) + \left( \mathbb{E}_{\boldsymbol{x}, \varepsilon(\boldsymbol{T}_\mathrm{X} \boldsymbol{x})} \big\| \boldsymbol{W}'^\top \underbrace{\boldsymbol{T}_\mathrm{W}^\top}_{=\, \boldsymbol{T}_\mathrm{X}^{-1}} \varepsilon \left( \boldsymbol{T}_\mathrm{X} \boldsymbol{x} \right) \big\|^2 \right) \\
&= \left( \mathbb{E}_{\boldsymbol{w}, \varepsilon(\boldsymbol{T}_\mathrm{W} \boldsymbol{w})} \big\| \boldsymbol{X}'^\top \boldsymbol{T}_\mathrm{W}^{-1} \varepsilon \left( \boldsymbol{T}_\mathrm{W} \boldsymbol{w} \right) \big\|^2 \right) + \left( \mathbb{E}_{\boldsymbol{x}, \varepsilon(\boldsymbol{T}_\mathrm{X} \boldsymbol{x})} \big\| \boldsymbol{W}'^\top \boldsymbol{T}_\mathrm{X}^{-1} \varepsilon \left( \boldsymbol{T}_\mathrm{X} \boldsymbol{x} \right) \big\|^2 \right).
\end{aligned}
\tag{38}
$$

**Eq. (17).**

$$
\left.
\begin{array}{ll}
\left( \operatorname{tr} \left( \boldsymbol{S} \right) \right)^2 \geq \operatorname{tr} \left( \boldsymbol{S}^2 \right) & (\boldsymbol{S} \succ \boldsymbol{0}) \\
\operatorname{tr} \left( \boldsymbol{S}^2 \right) \geq d^{-1} \left( \operatorname{tr} \left( \boldsymbol{S} \right) \right)^2 & \text{(Jensen's inequality)}
\end{array}
\right\} \Rightarrow \operatorname{tr} \left( \boldsymbol{S}^2 \right) \geq d^{-1} \left( \operatorname{tr} \left( \boldsymbol{S} \right) \right)^2 \geq d^{-1} \operatorname{tr} \left( \boldsymbol{S}^2 \right).
\tag{39}
$$

**Eq. (19).**

$$
\begin{aligned}
&\mathbb{E}_{\boldsymbol{y},\varepsilon(\boldsymbol{Ty})}\left\|\boldsymbol{T}^{-1}\varepsilon\left(\boldsymbol{Ty}\right)\right\|^2 \\
&= \mathbb{E}_{\boldsymbol{y},\boldsymbol{\eta}}\left\|\boldsymbol{T}^{-1}\operatorname{diag}\left(\boldsymbol{\eta}\right)\boldsymbol{Ty}\right\|^2 && \text{(Eq. (18))}\\
&= \mathbb{E}_{\boldsymbol{y},\boldsymbol{\eta}}\boldsymbol{y}^\top\boldsymbol{T}^\top\operatorname{diag}\left(\boldsymbol{\eta}\right)\boldsymbol{T}^{-\top}\boldsymbol{T}^{-1}\operatorname{diag}\left(\boldsymbol{\eta}\right)\boldsymbol{Ty} \\
&= \mathbb{E}_{\boldsymbol{y},\boldsymbol{\eta}}\operatorname{tr}\left(\boldsymbol{y}^\top\boldsymbol{T}^\top\operatorname{diag}\left(\boldsymbol{\eta}\right)\boldsymbol{T}^{-\top}\boldsymbol{T}^{-1}\operatorname{diag}\left(\boldsymbol{\eta}\right)\boldsymbol{Ty}\right) \\
&= \mathbb{E}_{\boldsymbol{y},\boldsymbol{\eta}}\operatorname{tr}\left(\boldsymbol{T}^{-1}\operatorname{diag}\left(\boldsymbol{\eta}\right)\boldsymbol{Tyy}^\top\boldsymbol{T}^\top\operatorname{diag}\left(\boldsymbol{\eta}\right)\boldsymbol{T}^{-\top}\right) && \text{(40)}\\
&= \operatorname{tr}\left(\boldsymbol{T}^{-1}\left(\mathbb{E}_{\boldsymbol{\eta}}\operatorname{diag}\left(\boldsymbol{\eta}\right)\boldsymbol{T}\left(\mathbb{E}_{\boldsymbol{y}}\boldsymbol{yy}^\top\right)\boldsymbol{T}^\top\operatorname{diag}\left(\boldsymbol{\eta}\right)\right)\boldsymbol{T}^{-\top}\right) \\
&= \operatorname{tr}\left(\boldsymbol{T}^{-1}\left(\mathbb{E}_{\boldsymbol{\eta}}\operatorname{diag}\left(\boldsymbol{\eta}\right)\boldsymbol{U}'\boldsymbol{S}'^2\boldsymbol{U}'^\top\operatorname{diag}\left(\boldsymbol{\eta}\right)\right)\boldsymbol{T}^{-\top}\right) && \text{(Eq. (37))}\\
&= \operatorname{tr}\left(\boldsymbol{T}^{-1}\left(\left(\mathbb{E}_{\boldsymbol{\eta}}\eta^2\right)\left(\boldsymbol{U}'\boldsymbol{S}'^2\boldsymbol{U}'^\top\right)\odot\mathbf{I}\right)\boldsymbol{T}^{-\top}\right) \\
&= \left(\mathbb{E}_{\boldsymbol{\eta}}\eta^2\right)\operatorname{tr}\left(\boldsymbol{T}^{-1}\left(\left(\boldsymbol{U}'\boldsymbol{S}'^2\boldsymbol{U}'^\top\right)\odot\mathbf{I}\right)\boldsymbol{T}^{-\top}\right).
\end{aligned}
$$

**Eq. (20).**

$$
\begin{aligned}
&\operatorname{tr}\left(\boldsymbol{T}^{-1}\left(\left(\boldsymbol{U}'\boldsymbol{S}'^2\boldsymbol{U}'^\top\right)\odot\mathbf{I}\right)\boldsymbol{T}^{-\top}\right) \\
&= \operatorname{tr}\left(\boldsymbol{T}^{-1}\operatorname{diag}\left(\left\|\boldsymbol{S}'\boldsymbol{U}'^\top\mathbf{e}_1\right\|^2,\ldots,\left\|\boldsymbol{S}'\boldsymbol{U}'^\top\mathbf{e}_d\right\|^2\right)\boldsymbol{T}^{-\top}\right) \\
&= \sum_{j=1}^{d}\sum_{k=1}^{d}\left(\mathbf{e}_j\boldsymbol{T}^{-1}\mathbf{e}_k\right)^2\left\|\boldsymbol{S}'\boldsymbol{U}'^\top\mathbf{e}_k\right\|^2 \\
&= \sum_{k=1}^{d}\left\|\boldsymbol{T}^{-1}\mathbf{e}_k\right\|^2\left\|\boldsymbol{S}'\boldsymbol{U}'^\top\mathbf{e}_k\right\|^2 \\
&= \sum_{k=1}^{d}\left\|\boldsymbol{USRS}'^{-1}\boldsymbol{U}'^\top\mathbf{e}_k\right\|^2\left\|\boldsymbol{S}'\boldsymbol{U}'^\top\mathbf{e}_k\right\|^2 && \text{(Eq. (16))}\\
&= \sum_{k=1}^{d}\left\|\boldsymbol{SRS}'^{-1}\boldsymbol{U}'^\top\mathbf{e}_k\right\|^2\left\|\boldsymbol{RS}'\boldsymbol{U}'^\top\mathbf{e}_k\right\|^2 && \text{(rotation-invariance of } L_2 \text{ norm)} \quad \text{(41)}\\
&\geq \sum_{k=1}^{d}\left(\mathbf{e}_k^\top\boldsymbol{U}'\boldsymbol{S}'^{-1}\boldsymbol{R}^\top\boldsymbol{SRS}'\boldsymbol{U}'^\top\mathbf{e}_k\right)^2 && \text{(Cauchy-Schwarz)}\\
&= d\left(d^{-1}\sum_{k=1}^{d}\left(\mathbf{e}_k^\top\boldsymbol{U}'\boldsymbol{S}'^{-1}\boldsymbol{R}^\top\boldsymbol{SRS}'\boldsymbol{U}'^\top\mathbf{e}_k\right)^2\right) \\
&\geq d\left(d^{-1}\sum_{k=1}^{d}\mathbf{e}_k^\top\boldsymbol{U}'\boldsymbol{S}'^{-1}\boldsymbol{R}^\top\boldsymbol{SRS}'\boldsymbol{U}'^\top\mathbf{e}_k\right)^2 && \text{(Jensen's inequality)}\\
&= d^{-1}\left(\operatorname{tr}\left(\boldsymbol{U}'\boldsymbol{S}'^{-1}\boldsymbol{R}^\top\boldsymbol{SRS}'\boldsymbol{U}'^\top\right)\right)^2 \\
&= d^{-1}\left(\operatorname{tr}\left(\boldsymbol{S}\right)\right)^2.
\end{aligned}
$$

The equality can be attained by choosing $U' = H$, $S' = S^{\frac{1}{2}}$, and $R = I$ such that

$$
\begin{aligned}
&\operatorname{tr}\left(T^{-1}\left(\left(U'S'^2U'^\top\right)\odot I\right)T^{-\top}\right) \\
&= \sum_{k=1}^d \left\|SRS'^{-1}U'^\top e_k\right\|^2 \left\|RS'U'^\top e_k\right\|^2 \quad \text{(Eq. (41))} \\
&= \sum_{k=1}^d \left(\left\|S^{\frac{1}{2}}H^\top e_k\right\|^2\right)^2 \qquad\qquad\qquad \text{(plug in)} \\
&= \sum_{k=1}^d \left(\sum_{j=1}^d s_j \left(e_j^\top H^\top e_k\right)^2\right)^2 \\
&= \sum_{k=1}^d \left(\sum_{j=1}^d s_j d^{-1}\right)^2 \\
&= d\left(d^{-1}\sum_{j=1}^d s_j\right)^2 \\
&= d^{-1}\left(\operatorname{tr}(S)\right)^2 .
\end{aligned}
\tag{42}
$$

**Eq. (21).**

$$
\begin{aligned}
&\mathbb{E}_{y,\varepsilon(Ty)}\left\|T^{-1}\varepsilon(Ty)\right\|^2 \\
&= \mathbb{E}_{y,\eta}\left\|T^{-1}\operatorname{diag}(\eta)Ty\right\|^2 \qquad\qquad\qquad\qquad \text{(Eq. (18))} \\
&= \mathbb{E}_{y,\eta}y^\top T^\top \operatorname{diag}(\eta)T^{-\top}T^{-1}\operatorname{diag}(\eta)Ty \\
&= \mathbb{E}_{y,\eta}y^\top T^\top \operatorname{diag}(\eta)^2 Ty \qquad\qquad\qquad \text{(orthogonality of } T) \\
&= \mathbb{E}_y y^\top T^\top \left(\mathbb{E}_\eta \operatorname{diag}(\eta)^2\right)Ty \\
&= \left(\mathbb{E}_\eta \eta^2\right)\mathbb{E}_y y^\top T^\top Ty \\
&= \left(\mathbb{E}_\eta \eta^2\right)\mathbb{E}_y y^\top y \qquad\qquad\qquad\qquad\quad \text{(orthogonality of } T) \\
&= \left(\mathbb{E}_\eta \eta^2\right)\mathbb{E}_y \|y\|^2 \\
&= \left(\mathbb{E}_\eta \eta^2\right)\operatorname{tr}\left(S^2\right) \qquad\qquad\qquad\qquad\quad \text{(Eq. (37))}.
\end{aligned}
\tag{43}
$$

**Eq. (22).**

In the case of $U' = I$:

$$
\begin{aligned}
&\operatorname{tr}\left(T^{-1}\left(\left(U'S'^2U'^\top\right)\odot I\right)T^{-\top}\right) \\
&= \operatorname{tr}\left(USRS'^{-1}U'^\top \left(\left(U'S'^2U'^\top\right)\odot I\right)U'S'^{-1}R^\top SU^\top\right) \quad \text{(Eq. (16))} \\
&= \operatorname{tr}\left(USRS'^{-1}\left(S'^2\odot I\right)S'^{-1}R^\top SU^\top\right) \qquad\qquad \text{(plug in)} \\
&= \operatorname{tr}\left(USRS'^{-1}S'^2 S'^{-1}R^\top SU^\top\right) \\
&= \operatorname{tr}\left(USRR^\top SU^\top\right) \\
&= \operatorname{tr}\left(US^2U^\top\right) \\
&= \operatorname{tr}\left(S^2\right) .
\end{aligned}
\tag{44}
$$

In the case of $S' = I$:

$$
\begin{aligned}
&\operatorname{tr}\left(T^{-1}\left(\left(U'S'^2U'^\top\right)\odot I\right)T^{-\top}\right)\\
={}&\operatorname{tr}\left(USRS'^{-1}U'^\top\left(\left(U'S'^2U'^\top\right)\odot I\right)U'S'^{-1}R^\top SU^\top\right)\quad\text{(Eq. (16))}\\
={}&\operatorname{tr}\left(USRU'^\top\left(\left(U'U'^\top\right)\odot I\right)U'R^\top SU^\top\right)\qquad\qquad\text{(plug in)}\\
={}&\operatorname{tr}\left(USRU'^\top U'R^\top SU^\top\right)\\
={}&\operatorname{tr}\left(US^2U^\top\right)\\
={}&\operatorname{tr}\left(S^2\right).
\end{aligned}
\tag{45}
$$

**Eq. (24).**

$$
\begin{aligned}
&\mathbb{E}_{y,\varepsilon(Ty)}\left\|T^{-1}\varepsilon\left(Ty\right)\right\|^2\\
={}&\mathbb{E}_{y,\eta}\left\|T^{-1}\left\|Ty\right\|_\infty\eta\right\|^2\qquad\qquad\text{(Eq. (23))}\\
={}&\mathbb{E}_{y,\eta}\left\|Ty\right\|_\infty^2\eta^\top T^{-\top}T^{-1}\eta\\
={}&\mathbb{E}_{y,\eta}\left\|Ty\right\|_\infty^2\operatorname{tr}\left(\eta^\top T^{-\top}T^{-1}\eta\right)\\
={}&\mathbb{E}_{y,\eta}\left\|Ty\right\|_\infty^2\operatorname{tr}\left(T^{-1}\eta\eta^\top T^{-\top}\right)\\
={}&\operatorname{tr}\left(T^{-1}\left(\mathbb{E}_\eta\eta\eta^\top\right)T^{-\top}\right)\mathbb{E}_y\left\|Ty\right\|_\infty^2\\
={}&\left(\mathbb{E}_\eta\eta^2\right)\operatorname{tr}\left(T^{-1}T^{-\top}\right)\mathbb{E}_y\left\|Ty\right\|_\infty^2\\
={}&\left(\mathbb{E}_\eta\eta^2\right)\left\|T^{-1}\right\|_F^2\mathbb{E}_y\left\|Ty\right\|_\infty^2.
\end{aligned}
\tag{46}
$$

**Eq. (25).**

$$
\begin{aligned}
\left\|T^{-1}\right\|_F^2&=\operatorname{tr}\left(T^{-1}T^{-\top}\right)\\
&=\operatorname{tr}\left(USRS'^{-1}U'^\top U'S'^{-1}R^\top SU^\top\right)\quad\text{(Eq. (16)}\\
&=\operatorname{tr}\left(SU^\top USRS'^{-1}U'^\top U'S'^{-1}R^\top\right)\\
&=\operatorname{tr}\left(S^2RS'^{-2}R^\top\right)
\end{aligned}
\tag{47}
$$

Because $s_1^2 \geq \cdots \geq s_d^2$ and $s_1'^{-2} \leq \cdots \leq s_d'^{-2}$, using von Neumann's trace inequality,

$$
\left\|T^{-1}\right\|_F^2 = \operatorname{tr}\left(S^2RS'^{-2}R^\top\right) \geq \operatorname{tr}\left(S^2S'^{-2}\right).
\tag{48}
$$

The equality can be attained by choosing $R = I$.

**Eq. (26).**

Express the expectations as

$$
\mathbb{E}_y\left\|Ty\right\|_\infty^2 = \mathbb{E}_y\max_k\left(e_k^\top Ty\right)^2 \geq \max_k\mathbb{E}_y\left(e_k^\top Ty\right)^2\quad\text{(Jensen's inequality)}
\tag{49}
$$

and

$$
\mathbb{E}_y\left\|Ty\right\|^2 = \mathbb{E}_y\sum_{k=1}^d\left(e_k^\top Ty\right)^2 = \sum_{k=1}^d\mathbb{E}_y\left(e_k^\top Ty\right)^2.
\tag{50}
$$

By the relationships among the mean, maximum, and summation values, we have:

$$
d^{-1}\mathbb{E}_y\left\|Ty\right\|^2 \leq \max_k\mathbb{E}_y\left(e_k^\top Ty\right)^2 \leq \mathbb{E}_y\left\|Ty\right\|_\infty^2 \leq \mathbb{E}_y\left\|Ty\right\|^2 \leq d\max_k\mathbb{E}_y\left(e_k^\top Ty\right)^2,
\tag{51}
$$

where, by Eq. (37),

$$
\mathbb{E}_y\left\|Ty\right\|^2 = \operatorname{tr}\left(S'^2\right)
\tag{52}
$$

and

$$
\mathbb{E}_y\left(e_k^\top Ty\right)^2 = e_k^\top T\left(\mathbb{E}_y yy^\top\right)T^\top e_k = e_k^\top U'S'^2U'^\top e_k = \left\|S'U'^\top e_k\right\|^2.
\tag{53}
$$

**Lemma A.2** (Maximum Inequalities for Tail-Bounded Distributions). *If $X_1, \ldots, X_d \in \mathbb{R}$ are zero-mean Gaussian random variables,*

$$\mathbb{E} \max_k X_k^2 \leq \min \{d, (2 \ln (2d) + 2)\} \max_k \mathbb{E} X_k^2. \tag{54}$$

*If $X_1, \ldots, X_d \in \mathbb{R}$ are zero-mean Laplacian random variables,*

$$\mathbb{E} \max_k X_k^2 \leq \left(2^{-1} (\ln d)^2 + \ln d + 1\right) \max_k \mathbb{E} X_k^2. \tag{55}$$

*Both inequalities hold without the need for independence.*

*Proof.* We provide proofs for the Gaussian and Laplacian distributions in Sections A.6.1 and A.6.2, respectively. □

Using Lemma A.2, for a tail-bounded $\mathcal{D}_y$ that is a zero-mean multivariate Gaussian or Laplacian, we can further tighten the upper bound by

$$\mathbb{E}_{\boldsymbol{y}} \|\boldsymbol{T}\boldsymbol{y}\|_\infty^2 \leq \min \left\{ d^{o(1)} \max_k \mathbb{E}_{\boldsymbol{y}} \left(\mathbf{e}_k^\top \boldsymbol{T}\boldsymbol{y}\right)^2, \mathbb{E}_{\boldsymbol{y}} \|\boldsymbol{T}\boldsymbol{y}\|^2 \right\}. \tag{56}$$

When $\boldsymbol{U}' = \boldsymbol{H}$, using Eqs. (37), (53), the marginal second moment in Eq. (56) becomes

$$\mathbb{E}_{\boldsymbol{y}} \left(\mathbf{e}_k^\top \boldsymbol{T}\boldsymbol{y}\right)^2 = \left\|\boldsymbol{S}'\boldsymbol{U}'^\top \mathbf{e}_k\right\|^2 = \left\|\boldsymbol{S}'\boldsymbol{H}^\top \mathbf{e}_k\right\|^2 = \sum_{j=1}^d s_j'^2 d^{-1} = d^{-1} \operatorname{tr} \left(\boldsymbol{S}'^2\right) = d^{-1} \mathbb{E}_{\boldsymbol{y}} \|\boldsymbol{T}\boldsymbol{y}\|^2, \tag{57}$$

which is equalized for all $k$, and $\max_k \mathbb{E}_{\boldsymbol{y}} \left(\mathbf{e}_k^\top \boldsymbol{T}\boldsymbol{y}\right)^2$ is minimized to $d^{-1} \mathbb{E}_{\boldsymbol{y}} \|\boldsymbol{T}\boldsymbol{y}\|^2$. Finally, combining Eqs. (51), (56), (57),

$$d^{-1} \operatorname{tr} \left(\boldsymbol{S}'^2\right) \leq \mathbb{E}_{\boldsymbol{y}} \|\boldsymbol{T}\boldsymbol{y}\|_\infty^2 \leq \begin{cases} d^{o(1)-1} \operatorname{tr} \left(\boldsymbol{S}'^2\right), & \text{tail-bounded } \mathcal{D}_y, \\ \operatorname{tr} \left(\boldsymbol{S}'^2\right), & \text{otherwise.} \end{cases} \tag{58}$$

**Eq. (31).**

For orthogonal $\boldsymbol{T}$, we have $\left\|\boldsymbol{T}^{-1}\right\|_F^2 = d$ and $\mathbb{E}_{\boldsymbol{y}} \|\boldsymbol{T}\boldsymbol{y}\|^2 = \mathbb{E}_{\boldsymbol{y}} \|\boldsymbol{y}\|^2$. Using Eq. (37), $\mathbb{E}_{\boldsymbol{y}} \|\boldsymbol{y}\|^2 = \operatorname{tr} \left(\boldsymbol{S}^2\right)$. By Eq. (51),

$$\operatorname{tr} \left(\boldsymbol{S}^2\right) = d^{-1} \left\|\boldsymbol{T}^{-1}\right\|_F^2 \mathbb{E}_{\boldsymbol{y}} \|\boldsymbol{T}\boldsymbol{y}\|^2 \leq \left\|\boldsymbol{T}^{-1}\right\|_F^2 \mathbb{E}_{\boldsymbol{y}} \|\boldsymbol{T}\boldsymbol{y}\|_\infty^2 \leq \left\|\boldsymbol{T}^{-1}\right\|_F^2 \mathbb{E}_{\boldsymbol{y}} \|\boldsymbol{T}\boldsymbol{y}\|^2 = d \operatorname{tr} \left(\boldsymbol{S}^2\right). \tag{59}$$

## A.5. Quantization Error Modeling Justifications

### A.5.1. INTEGER (INT) TYPES

We model the quantization error using pseudo noise similar to DiffQ (Défossez et al., 2022).

Define the scalar quantizer of $b$-bit symmetric integer as

$$q(x) = \left(\left\lfloor xs^{-1} \right\rfloor + 2^{-1}\right) s, \tag{60}$$

where $x \in \mathbb{R}$, scale $s \in \mathbb{R}_{\neq 0}$, and $xs^{-1} \in \left[-2^{b-1}, 2^{b-1}\right)$.

To simplify the analysis, we model the quantizer as

$$q(x) = \left(xs^{-1} + \xi\right) s = x + s\xi \tag{61}$$

with random noise $\xi \sim \operatorname{Uniform} \left(-2^{-1}, 2^{-1}\right)$. Then, the quantization error can be expressed as

$$q(x) - x = s\xi. \tag{62}$$

For a vector $\boldsymbol{\alpha} \in \mathbb{R}^d$, we use the AbsMax scale $s = 2^{1-b} \|\boldsymbol{\alpha}\|_\infty$. Define

$$\eta = 2^{1-b} \xi, \tag{63}$$

then we have $s\xi = \eta \|\boldsymbol{\alpha}\|_\infty$. The quantization error of vector $\boldsymbol{\alpha}$ is

$$\varepsilon(\boldsymbol{\alpha}) = \|\boldsymbol{\alpha}\|_\infty \boldsymbol{\eta} \tag{64}$$

with $\boldsymbol{\eta} = [\eta_1, \ldots, \eta_d]^\top \in \mathbb{R}^d$ being a random vector of i.i.d. samples $\eta \sim \mathcal{D}_\eta$ and

$$\mathbb{E}_\eta \eta = 2^{1-b} \mathbb{E}_\xi \xi = 0 \tag{65}$$

### A.5.2. FLOATING-POINT (FP) TYPES

The FP format E$a$M$b$ is defined as a tuple $(s, e, m)$ with a sign $s \in \{0, 1\}$, an exponent $e \in \{0, \ldots, 2^a - 1\}$, and a mantissa $m \in \{0, \ldots, 2^b - 1\}$. Assume we do not need to consider the subnormal and special number cases. The tuple $(s, e, m)$ represents the number

$$x_{\text{E}a\text{M}b} = (-1)^s \, 2^{e - 2^{a-1} + 1} \left(1 + 2^{-b} m\right). \tag{66}$$

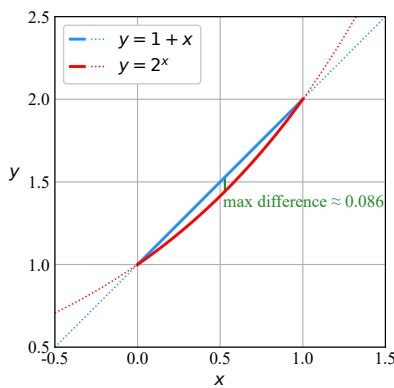

*Figure 6.* The two curves $y = 1 + x$ and $y = 2^x$ are close for $0 \le x \le 1$.

Using the approximation $1 + (\cdot) \approx 2^{(\cdot)}$ for $2^{-b} m \in [0, 1]$ (visualized in Figure 6), we define a smoothed version of E$a$M$b$, named SE$a$M$b$, as

$$x_{\text{SE}a\text{M}b} = (-1)^s \, 2^{e - 2^{a-1} + 1} 2^{2^{-b} m} = (-1)^s \, 2^{2^{-b}(2^b e + m) - 2^{a-1} + 1} \tag{67}$$

with

$$2^b e + m = 2^b \left(\log_2 |x_{\text{SE}a\text{M}b}| + 2^{a-1} - 1\right) \tag{68}$$

being an $(a + b)$-bit unsigned integer (concatenating the binary representations of $e$ and $m$). The minimum and maximum of $|x_{\text{SE}a\text{M}b}|$ are

$$\begin{aligned}
\min |x_{\text{SE}a\text{M}b}| &= 2^{-2^{a-1} + 1}, \\
\max |x_{\text{SE}a\text{M}b}| &= 2^{2^{-b}(2^{a+b} - 1) - 2^{a-1} + 1} = 2^{2^{a-1} - 2^{-b} + 1},
\end{aligned} \tag{69}$$

respectively.

Denote SE$a$M$b$ $(x)$ as the quantizer that casts $x \in [-\max |x_{\text{SE}a\text{M}b}|, \max |x_{\text{SE}a\text{M}b}|]$ to the nearest number in the SE$a$M$b$ type. Assume $e \in (-\infty, 2^a - 1]$ so that the smaller values can be represented more precisely. SE$a$M$b$ $(x)$ can be approximated using integer rounding in logarithmic space and first-order expansion. For $x \neq 0$,

$$\begin{aligned}
\text{SE}a\text{M}b\,(x) &\approx \text{sgn}\,(x)\, 2^{2^{-b}\left\lfloor 2^b\left(\log_2 |x| + 2^{a-1} - 1\right)\right\rceil - 2^{a-1} + 1} \\
&= \text{sgn}\,(x)\, 2^{2^{-b}\left\lfloor 2^b \log_2 |x|\right\rceil} \\
&\approx \text{sgn}\,(x) \left(2^{2^{-b} 2^b \log_2 |x|} + \left(\left\lfloor 2^b \log_2 |x|\right\rceil - 2^b \log_2 |x|\right) 2^{2^{-b} 2^b \log_2 |x|} 2^{-b} \ln 2\right) \\
&= x \left(1 + \left(\left\lfloor 2^b \log_2 |x|\right\rceil - 2^b \log_2 |x|\right) 2^{-b} \ln 2\right).
\end{aligned} \tag{70}$$

For $x = 0$,

$$\text{SE}a\text{M}b\,(0) \approx \lim_{x \to 0} x \left(1 + \left(\left\lfloor 2^b \log_2 |x|\right\rceil - 2^b \log_2 |x|\right) 2^{-b} \ln 2\right) = 0. \tag{71}$$

We model the casting error as

$$\text{SE}a\text{M}b\,(x) - x = (\ln 2)\, 2^{-b} \xi x \tag{72}$$

where $x \in [-\max |x_{\text{SE}a\text{M}b}|, \max |x_{\text{SE}a\text{M}b}|]$ and, similar to the INT types in Eq. (61), the random variable $\xi \sim \text{Uniform}\left(-2^{-1}, 2^{-1}\right)$.

For a quantization format that uses $\mathrm{E}a\mathrm{M}b$ for values and $\mathrm{E}\tilde{a}\mathrm{M}\tilde{b}$ for scales, we approximate it with a scalar quantizer $q(x) = \mathrm{SE}a\mathrm{M}b\left(xs^{-1}\right)\mathrm{SE}\tilde{a}\mathrm{M}\tilde{b}(s)$ where $x \in \mathbb{R}$, scale $s \in \mathbb{R}_{\neq 0}$, $|s| \leq \max\left|x_{\mathrm{SE}\tilde{a}\mathrm{M}\tilde{b}}\right|$, and $\left|xs^{-1}\right| \leq \max\left|x_{\mathrm{SE}a\mathrm{M}b}\right|$. Then,

$$
\begin{aligned}
q(x) &= \mathrm{SE}a\mathrm{M}b\left(xs^{-1}\right)\mathrm{SE}\tilde{a}\mathrm{M}\tilde{b}(s) \\
&= \left(1 + (\ln 2)\, 2^{-b}\xi\right)xs^{-1}\left(1 + (\ln 2)\, 2^{-\tilde{b}}\tilde{\xi}\right)s \\
&= \left(1 + (\ln 2)\, 2^{-b}\xi\right)\left(1 + (\ln 2)\, 2^{-\tilde{b}}\tilde{\xi}\right)x
\end{aligned}
\tag{73}
$$

with the independent random variables $\xi, \tilde{\xi} \sim \mathrm{Uniform}\left(-2^{-1}, 2^{-1}\right)$.

Define

$$
\eta = \left(1 + (\ln 2)\, 2^{-b}\xi\right)\left(1 + (\ln 2)\, 2^{-\tilde{b}}\tilde{\xi}\right) - 1.
\tag{74}
$$

Then, the quantization error can be expressed as

$$
q(x) - x = \eta x.
\tag{75}
$$

And the expected quantization error is 0 because

$$
\mathbb{E}_\eta \eta = \mathbb{E}_{\xi, \tilde{\xi}}\, (\ln 2)\, 2^{-b}\xi + (\ln 2)\, 2^{-\tilde{b}}\tilde{\xi} + (\ln 2)^2\, 2^{-b-\tilde{b}}\xi\tilde{\xi} = 0.
\tag{76}
$$

### A.6. Maximum Inequalities for Tail-Bounded Distributions

**Lemma A.3** (Expectation from the Survival Function). *Let $X \geq 0$ be a random variable with the density function $f$ and the survival function*

$$
S(t) = \mathbb{P}(X \geq t)
\tag{77}
$$

*with $t \geq 0$. Let $g : [0, \infty) \to \mathbb{R}$ be integrable and define*

$$
G(t) = \int_0^t g(u)\, \mathrm{d}u.
\tag{78}
$$

*Assume that*

$$
\lim_{t \to \infty} G(t)\, S(t) = 0.
\tag{79}
$$

*Then,*

$$
\mathbb{E}G(X) = \int_0^\infty g(t)\, S(t)\, \mathrm{d}t.
\tag{80}
$$

*Proof.* Since $X$ has density $f$ and takes values in $[0, \infty)$,

$$
\mathbb{E}G(X) = \int_0^\infty G(t)\, f(t)\, \mathrm{d}t.
\tag{81}
$$

The survival function satisfies

$$
S(t) = \mathbb{P}(X \geq t) = \int_t^\infty f(u)\, \mathrm{d}u,
\tag{82}
$$

so $S$ is differentiable almost everywhere with $S'(t) = -f(t)$. Hence,

$$
\mathbb{E}G(X) = \int_0^\infty G(t)\, f(t)\, \mathrm{d}t = -\int_0^\infty G(t)\, S'(t)\, \mathrm{d}t.
\tag{83}
$$

We now integrate by parts, using $G'(t) = g(t)$,

$$
-\int_0^\infty G(t)\, S'(t)\, \mathrm{d}t = -\left[G(t)\, S(t)\right]_0^\infty + \int_0^\infty g(t)\, S(t)\, \mathrm{d}t.
\tag{84}
$$

By definition, $G(0) = 0$, and by assumption $\lim_{t \to \infty} G(t)\, S(t) = 0$, the boundary term $\left[G(t)\, S(t)\right]_0^\infty = 0$. Therefore,

$$
\mathbb{E}G(X) = \int_0^\infty g(t)\, S(t)\, \mathrm{d}t.
\tag{85}
$$

$\square$

A.6.1. GAUSSIAN DISTRIBUTION

**Definition of sub-Gaussian.**

A probability distribution of a random variable $X \in \mathbb{R}$ is called sub-Gaussian with variance proxy $\sigma^2$ if

$$\mathbb{E} \exp\left((X - \mathbb{E}X)\, t\right) \leq \exp\left(2^{-1}\sigma^2 t^2\right) \tag{86}$$

for all $t \in \mathbb{R}$, and the smallest such $\sigma^2$ is called the optimal variance proxy.

**Basic properties of sub-Gaussian distributions.**

The variance proxy is larger than or equal to the variance: $\sigma^2 \geq \mathbb{E}\left(X - \mathbb{E}X\right)^2$. For a Gaussian distribution, the optimal variance proxy is the same as the variance.

Chernoff bound on the tail:

$$\mathbb{P}\left(|X - \mathbb{E}X| \geq t\right) \leq 2 \exp\left(-2^{-1}\sigma^{-2}t^2\right) \tag{87}$$

for all $t \geq 0$.

**Maximum inequality.**

Let $X_1, \ldots, X_d$ be zero-mean Gaussian random variables with variance $\sigma_1^2, \ldots, \sigma_d^2$. Denote

$$\sigma_\star^2 = \max_k \sigma_k^2 = \max_k \mathbb{E}X_k^2 \tag{88}$$

as the smallest variance proxy for $X_1, \ldots, X_d$. Using Lemma A.3,

$$\begin{aligned}
\mathbb{E} \max_k X_k^2 &= 2 \int_0^\infty t\, \mathbb{P}\left(\max_k |X_k| \geq t\right) \mathrm{d}t \\
&= 2 \int_0^\infty t\, \mathbb{P}\left(\bigcup_{k=1}^d \{|X_k| \geq t\}\right) \mathrm{d}t \\
&\leq 2 \int_0^{t_\star} t\, 1\, \mathrm{d}t + 2 \int_{t_\star}^\infty t \left(\sum_{k=1}^d \mathbb{P}\left(|X_k| \geq t\right)\right) \mathrm{d}t \\
&\leq 2 \int_0^{t_\star} t\, \mathrm{d}t + 2d \int_{t_\star}^\infty t \left(\max_k \mathbb{P}\left(|X_k| \geq t\right)\right) \mathrm{d}t \\
&\leq 2 \int_0^{t_\star} t\, \mathrm{d}t + 4d \int_{t_\star}^\infty t \exp\left(-2^{-1}\sigma_\star^{-2}t^2\right) \mathrm{d}t \\
&= \left[t^2\right]_0^{t_\star} + 4d \left[-\sigma_\star^2 \exp\left(-2^{-1}\sigma_\star^{-2}t^2\right)\right]_{t_\star}^\infty \\
&= t_\star^2 + 4d\sigma_\star^2 \exp\left(-2^{-1}\sigma_\star^{-2}t_\star^2\right).
\end{aligned} \tag{89}$$

Splitting at $t_\star = \sigma_\star \sqrt{2\ln(2d)}$ gives

$$\mathbb{E} \max_k X_k^2 \leq 2\sigma_\star^2 \ln(2d) + 2\sigma_\star^2 = \left(2\ln(2d) + 2\right) \max_k \mathbb{E}X_k^2 \tag{90}$$

without needing the independence of $X_k$. Also, considering the trivial bound

$$\mathbb{E} \max_k X_k^2 \leq \mathbb{E} \sum_{k=1}^d X_k^2 = \sum_{k=1}^d \mathbb{E}X_k^2 \leq d \max_k \mathbb{E}X_k^2, \tag{91}$$

then,

$$\mathbb{E} \max_k X_k^2 \leq \min\left\{d, \left(2\ln(2d) + 2\right)\right\} \max_k \mathbb{E}X_k^2. \tag{92}$$

### A.6.2. LAPLACIAN DISTRIBUTION

For a random variable $X \sim \text{Laplace}\,(\mu, b)$ and $t \geq 0$,

$$\mathbb{P}\left(|X - \mu| \geq t\right) = 2 \int_t^\infty 2^{-1} b^{-1} \exp\left(-b^{-1} x\right) \mathrm{d}x = \exp\left(-b^{-1} t\right). \tag{93}$$

The variance $\mathbb{E}\left(X - \mathbb{E}X\right)^2 = 2b^2$.

Let $X_1, \ldots, X_d$ be zero-mean Laplacian random variables with scale parameters $b_1, \ldots, b_d$. Denote

$$b_\star = \max_k b_k = \max_k \sqrt{\frac{1}{2} \mathbb{E}X_k^2} = \sqrt{\frac{1}{2} \max_k \mathbb{E}X_k^2}. \tag{94}$$

For $t \geq 0$,

$$\mathbb{P}\left(|X_k| \geq t\right) = \exp\left(-b_k^{-1} t\right) \leq \exp\left(-b_\star^{-1} t\right). \tag{95}$$

Using Lemma A.3,

$$\begin{aligned}
\mathbb{E} \max_k X_k^2 &= 2 \int_0^\infty t\, \mathbb{P}\left(\max_k |X_k| \geq t\right) \mathrm{d}t \\
&= 2 \int_0^\infty t\, \mathbb{P}\left(\bigcup_{k=1}^d \{|X_k| \geq t\}\right) \mathrm{d}t \\
&\leq 2 \int_0^{t_\star} t\, 1 \,\mathrm{d}t + 2 \int_{t_\star}^\infty t\left(\sum_{k=1}^d \mathbb{P}\left(|X_k| \geq t\right)\right) \mathrm{d}t \\
&\leq 2 \int_0^{t_\star} t\, \mathrm{d}t + 2d \int_{t_\star}^\infty t\left(\max_k \mathbb{P}\left(|X_k| \geq t\right)\right) \mathrm{d}t \\
&\leq 2 \int_0^{t_\star} t\, \mathrm{d}t + 2d \int_{t_\star}^\infty t \exp\left(-b_\star^{-1} t\right) \mathrm{d}t \\
&= \left[t^2\right]_0^{t_\star} + 2d \left[-b_\star \left(t + b_\star\right) \exp\left(-b_\star^{-1} t\right)\right]_{t_\star}^\infty \\
&= t_\star^2 + 2d b_\star \left(t_\star + b_\star\right) \exp\left(-b_\star^{-1} t_\star\right).
\end{aligned} \tag{96}$$

Splitting at $t_\star = b_\star \ln d$ gives

$$\mathbb{E} \max_k X_k^2 \leq b_\star^2 \left((\ln d)^2 + 2 \ln d + 2\right) = \left(2^{-1} (\ln d)^2 + \ln d + 1\right) \max_k \mathbb{E}X_k^2 \tag{97}$$

without needing the independence of $X_k$.

# B. Further Experimental Results

## B.1. Additional Accuracy Results

In this section, we report additional evaluations conducted across the Llama 3 and Qwen 3 family models in Tables 3 to 5 and Figures 7 to 10. For the Qwen models, the thinking mode is disabled across all experiments. The evaluation results are obtained using emulated ("fake") quantization, where quantized values are represented in the BF16/FP16 format. Results are aggregated over multiple random seeds.

*Table 3.* W4A4 accuracy results on the LM Evaluation Harness for different models with RTN, GPTQ quantizations, and identity (I), Hadamard (H), WUSH transforms. For the Llama model, the MMLU-CoT metric is reported in the MMLU column.

| Model | Format | Quantization | Transform | MMLU | GSM8K | HellaSwag | WinoGrande | Average | Recovery |
|---|---|---|---|---|---|---|---|---|---|
| Llama-3.2-3B-Instruct | BF16 | - | - | 64.43 | 78.01 | 73.42 | 70.09 | 71.49 | 100.0 |
| | NVFP4 | RTN | I | 61.07 | 72.01 | 70.90 | 66.77 | 67.69 | 94.68 |
| | | | H | 59.91 | 64.82 | 69.77 | 65.59 | 65.02 | 90.95 |
| | | | WUSH | 59.91 | 71.24 | 70.96 | 67.25 | 67.34 | 94.19 |
| | | GPTQ | H | 60.22 | 70.91 | 71.06 | 66.57 | 67.19 | 93.99 |
| | | | WUSH | 60.17 | 72.71 | 71.26 | 67.32 | **67.87** | **94.93** |
| | MXFP4 | RTN | I | 56.81 | 60.80 | 67.30 | 64.56 | 62.37 | 87.24 |
| | | | H | 55.99 | 61.11 | 68.36 | 64.09 | 62.39 | 87.27 |
| | | | WUSH | 59.54 | 69.49 | 70.20 | 66.47 | **66.43** | **92.92** |
| | | GPTQ | H | 60.24 | 68.84 | 69.58 | 66.08 | 66.18 | 92.58 |
| | | | WUSH | 59.39 | 68.26 | 69.88 | 67.07 | 66.15 | 92.53 |
| Qwen3-8B | BF16 | - | - | 72.98 | 90.90 | 75.52 | 70.56 | 77.49 | 100.0 |
| | NVFP4 | RTN | I | 70.47 | 88.40 | 74.44 | 69.53 | 75.71 | 97.70 |
| | | | H | 70.19 | 86.35 | 73.02 | 68.11 | 74.42 | 96.04 |
| | | | WUSH | 70.86 | 89.92 | 74.84 | 70.02 | 76.42 | 98.61 |
| | | GPTQ | H | 70.78 | 89.11 | 74.34 | 69.25 | 75.87 | 97.91 |
| | | | WUSH | 71.03 | 89.56 | 74.88 | 70.32 | **76.45** | **98.66** |
| | MXFP4 | RTN | I | 67.69 | 84.23 | 71.24 | 67.40 | 72.64 | 93.74 |
| | | | H | 67.53 | 87.04 | 71.48 | 67.64 | 73.42 | 94.75 |
| | | | WUSH | 69.96 | 86.90 | 73.87 | 68.92 | 74.91 | 96.68 |
| | | GPTQ | H | 69.58 | 88.28 | 73.75 | 69.25 | 75.22 | 97.07 |
| | | | WUSH | 70.40 | 88.80 | 74.36 | 69.19 | **75.69** | **97.68** |
| Qwen3-14B | BF16 | - | - | 77.18 | 91.96 | 79.84 | 74.27 | 80.81 | 100.0 |
| | NVFP4 | RTN | I | 75.20 | 90.22 | 77.38 | 73.32 | 79.03 | 97.80 |
| | | | H | 74.98 | 92.04 | 77.76 | 72.38 | 79.29 | 98.12 |
| | | | WUSH | 75.43 | 91.78 | 78.82 | 73.18 | **79.80** | **98.76** |
| | MXFP4 | RTN | I | 72.92 | 90.22 | 76.68 | 71.51 | 77.83 | 96.31 |
| | | | H | 73.13 | 87.41 | 76.29 | 71.67 | 77.13 | 95.44 |
| | | | WUSH | 74.56 | 91.87 | 78.18 | 73.15 | **79.44** | **98.30** |
| Qwen3-32B | BF16 | - | - | 80.74 | 92.12 | 83.96 | 76.95 | 83.44 | 100.0 |
| | NVFP4 | RTN | I | 78.92 | 90.14 | 82.60 | 76.64 | 82.08 | 98.37 |
| | | | H | 79.48 | 91.81 | 82.51 | 74.98 | 82.20 | 98.51 |
| | | | WUSH | 79.56 | 92.72 | 83.14 | 75.79 | **82.80** | **99.24** |
| | MXFP4 | RTN | I | 76.82 | 75.28 | 81.32 | 74.74 | 77.04 | 92.33 |
| | | | H | 78.63 | 92.57 | 81.87 | 75.53 | **82.15** | **98.45** |
| | | | WUSH | 78.70 | 90.83 | 82.63 | 75.87 | 82.01 | 98.28 |

*Table 4.* KL-divergence for Qwen3-8B measured on a slice of C4 dataset, with calibration performed on the FineWeb dataset. Results (lower is better) are averaged over multiple random seeds. H denotes the Hadamard transform.

| Model | Format | Quantization | Transform | KL |
|---|---|---|---|---|
| Qwen3-8B | NVFP4 | GPTQ | H | 0.069518 |
| | | RTN | WUSH | 0.065747 |
| | | GPTQ | WUSH | **0.054646** |
| | MXFP4 | GPTQ | H | 0.093119 |
| | | RTN | WUSH | 0.093739 |
| | | GPTQ | WUSH | **0.077909** |

*Table 5.* Sensitivity of WUSH to calibration datasets on Qwen3-8B with MXFP4.

| Quantization | Calibration Dataset | LM Eval Harness Average | Platinum Benchmarks Average |
|---|---|---|---|
| RTN | FineWeb | 74.91 | 93.00 |
| | C4 | 75.57 | 92.89 |
| | Open-Platypus | 74.99 | 93.12 |
| GPTQ | FineWeb | 75.69 | 93.52 |
| | C4 | 75.78 | 93.53 |

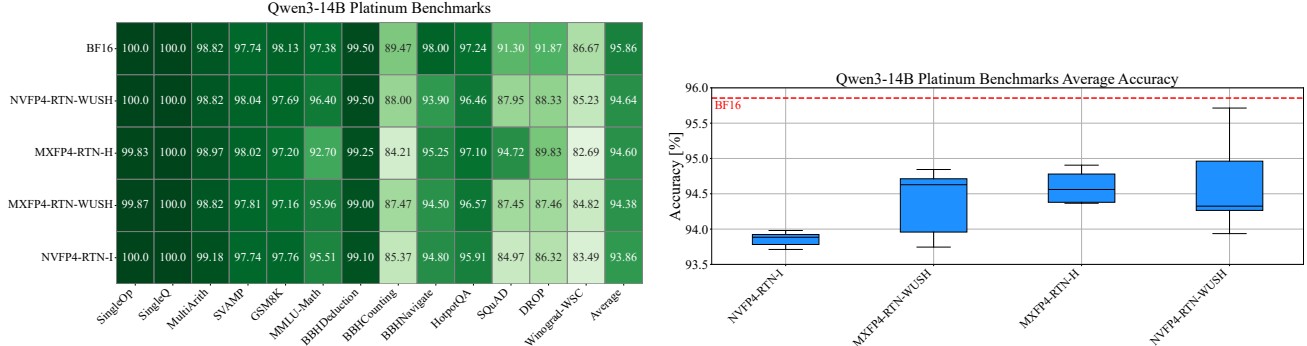

*Figure 7.* Comparison of different transforms on Qwen3-14B for both NVFP4 and MXFP4 quantization on Platinum Benchmarks. The left table shows accuracy results across the individual benchmark tasks, while the right plot shows the average accuracy scores together with their standard deviations for each transform.

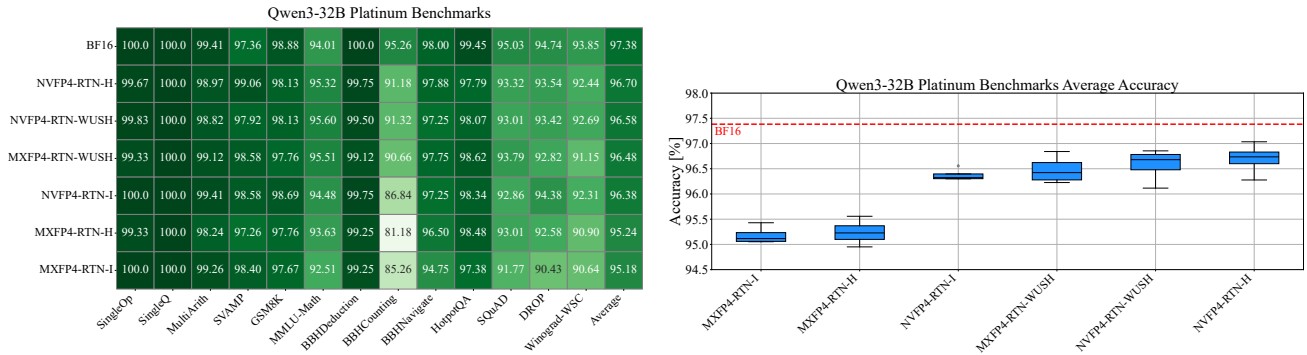

*Figure 8.* Comparison of different transforms on Qwen3-32B for both NVFP4 and MXFP4 quantization on Platinum Benchmarks. The left table shows accuracy results across the individual benchmark tasks, while the right plot shows the average accuracy scores together with their standard deviations for each transform.

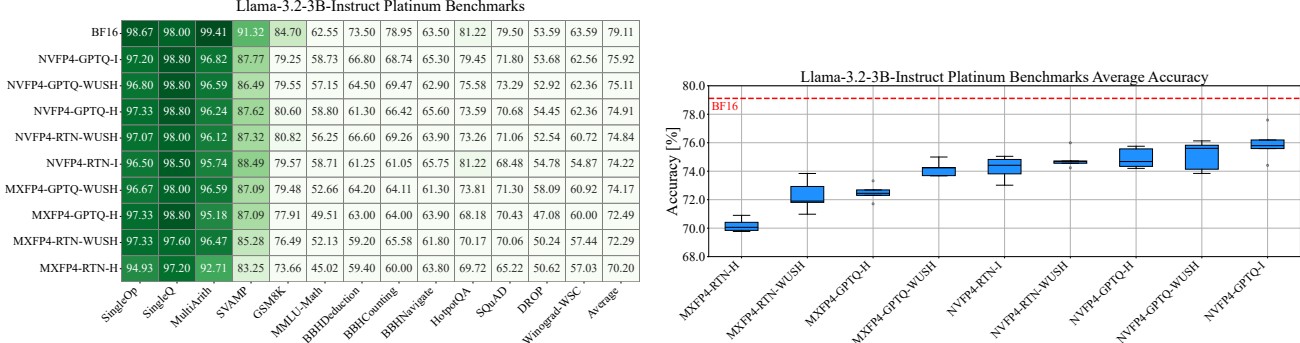

*Figure 9.* Comparison of different transforms on Llama-3.2-3B-Instruct for both NVFP4 and MXFP4 quantization on Platinum Benchmarks. The left table shows accuracy results across the individual benchmark tasks, while the right plot shows the average accuracy scores together with their standard deviations for each transform.

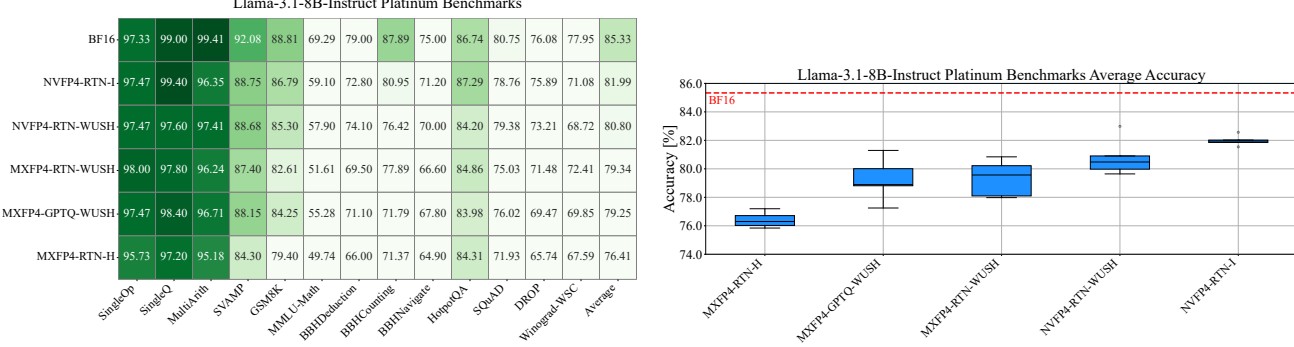

*Figure 10.* Comparison of different transforms on Llama-3.1-8B-Instruct for both NVFP4 and MXFP4 quantization on Platinum Benchmarks. The left table shows accuracy results across the individual benchmark tasks, while the right plot shows the average accuracy scores together with their standard deviations for each transform.

## B.2. Computational and Memory Costs

Table 6 reports the offline preprocessing cost of WUSH with RTN. These measurements include the calibration and transform construction pipeline. Note that the offline programs are generally not optimally implemented, and there are some configuration choices that can significantly vary the measurements. For example, one can choose to offload and recompute the calibration activations to save GPU memory at the cost of increased runtime.

Table 7 reports the additional whole-checkpoint storage overhead from storing the activation-side WUSH transforms.

*Table 6.* Offline preprocessing cost of WUSH with RTN.

| Model | GPU | Time [min] | GPU Memory [GB] |
|---|---|---|---|
| Llama-3.2-3B-Instruct | H100 | 9 | 10 |
| Llama-3.1-8B-Instruct | H100 | 19 | 19 |
| Qwen3-8B | H100 | 25 | 17 |
| Qwen3-14B | H100 | 30 | 20 |
| Qwen3-32B | B200 | 38 | 40 |

*Table 7.* Whole-checkpoint storage overhead from storing the activation-side WUSH transforms.

| Model | MXFP4 | NVFP4 |
|---|---|---|
| Llama-3.2-3B-Instruct | 2.1% | 1.0% |
| Llama-3.1-8B-Instruct | 1.4% | 0.7% |
| Qwen3-8B | 1.4% | 0.7% |
| Qwen3-14B | 1.2% | 0.6% |
| Qwen3-32B | 1.2% | 0.6% |

