# OpenReview forum: "WUSH: Near-Optimal Adaptive Transforms for LLM Quantization"
_ICML.cc/2026/Conference — ICML 2026 regular_

### Official Review · Reviewer_tLCu · 2026-03-02

**Soundness:** 3
**Presentation:** 3
**Significance:** 2
**Originality:** 3
**Overall Recommendation:** 4
**Confidence:** 4

**Summary:**

This paper addresses the degradation caused by outliers in weight and activation quantization for LLMs. The authors derive WUSH, a closed-form, data-aware linear blockwise transform, which they prove is optimal for floating-point (FP) quantizers and near-optimal for integer (INT) formats under specific distributional assumptions. WUSH integrates a Hadamard backbone with second-moment statistics to reshape data distributions, supported by a specialized GPU kernel that maintains the throughput of data-agnostic baselines while significantly improving accuracy in low-bit formats like MXFP4.

**Compliance With Llm Reviewing Policy:**

Affirmed.

**Final Justification:**

The rebuttal has addressed my main concerns. Hope to see the authors' promised revisions if accepeted.

**Key Questions For Authors:**

Please see weaknesses.

**Limitations:**

yes

**Strengths And Weaknesses:**

**Strengths**

1. The work provides a rigorous theoretical foundation for why Hadamard transforms are effective and how they can be extended via second-order statistics to reach optimality. Theorem 4.1 and the accompanying proofs offer a principled justification that moves beyond the heuristic approaches common in prior literature.
2. The derivation of a closed-form solution is a significant practical advantage, as it avoids the expensive iterative optimization or fine-tuning required by methods like SpinQuant or FlatQuant.
3. The systems contribution is robust; the authors do not merely propose a mathematical transform but also implement a fused GPU kernel. Figure 2 demonstrates that WUSH achieves up to 6.6x speedups over BF16, nearly matching the performance of less sophisticated Hadamard-only kernels.

**Weaknesses**

1. I have a concern regarding the statement in Lines 27–30 where the paper claims: *“As opposed to SpinQuant, our approach is closed-form and calibration-based.”* SpinQuant is described as data-unaware, but in practice, the rotations $(R_1, R_2)$ applied to $W$ in SpinQuant are obtained through calibration, similar to this paper. Only the SpinQuant + Hadamard setting introduces non-calibrated rotations $(R_3, R_4)$. Therefore, I find this comparison somewhat unclear.

2. The method requires storing the $(T_{\text{wush}(i)})$ matrices. Although the authors mention they are stored as $(G, G, C)$, the memory overhead of these transforms is not explicitly stated.  As a strong baseline in this paper, SpinQuant’s $(R_1, R_2)$ can be absorbed into the weight matrix $W$ itself. Please provide a clear comparison of the memory overhead for the experimental models in this work.

3. Group size has a significant impact on the performance of quantized models. In Table 2, what group sizes are used for WUSH and the other baselines?

4. Why does the “WUS” (without Hadamard) variant perform significantly worse than the identity transform in the INT4 setting (Table 1)?

5. The use of stochastic unbiased quantization in the theory (Section 4.1) while the actual implementation uses Round-To-Nearest (RTN). While standard in the field, this gap between theoretical assumptions and empirical execution remains a point of friction that the authors should discuss more explicitly.

---

> ### Author Rebuttal · Authors · 2026-03-30
>
> Thank you for your time and detailed feedback. Below, we address your concerns and questions.
>
> **W1: SpinQuant statement.**
>
> Our intended distinction was not calibration-based vs non-calibration-based, but rather closed-form vs learned or iteratively optimized (e.g., gradient descent). To avoid confusion, we will revise this sentence to say that WUSH is a closed-form calibration-driven transform, whereas SpinQuant/FlatQuant learn transforms on calibration data.
>
> **W2: Memory overhead.**
>
> We consider a layer with input size $d\_{\mathrm{in}}$, output size $d\_{\mathrm{out}}$, and transform/quantization block size $d$. The offline weight-side transform is absorbed into the transformed weights. The only additional stored object is the online activation transform, for which we keep only the diagonal blocks in $(G, G, C)$ format where the block size $G=d$ and the number of blocks $C=\frac{d\_{\mathrm{in}}}{d}$. Hence the transform storage per layer is $d\times d \times\frac{d\_{\mathrm{in}}}{d}=dd\_{\mathrm{in}}$, whereas the weight matrix has $d_{\mathrm{in}}d_{\mathrm{out}}$ elements. For 4-bit weight (plus groupwise 8-bit scale) and 16-bit transform, the overhead to store the transform is approximately $\frac{16dd_{\mathrm{in}}}{4d_{\mathrm{in}}d_{\mathrm{out}}}=\frac{4d}{d_{\mathrm{out}}}$. Since $d$ is 16 or 32, and $d\_{\mathrm{out}}$ is several thousand, the overhead is small. In the revision, we will add the resulting whole-checkpoint overheads for our experimental models, also listed below.
>
> |Model|MXFP4|NVFP4|
> |-|-|-|
> |LLama-3.2-3B-Instruct|2.1%|1.0%|
> |LLama-3.1-8B-Instruct|1.4%|0.7%|
> |Qwen3-8B|1.4%|0.7%|
> |Qwen3-14B|1.2%|0.6%|
> |Qwen3-32B|1.2%|0.6%|
>
> **W3: Group sizes in Table 2.**
>
> The quantization group size is fixed by the MXFP4/NVFP4 format definitions. MXFP4 always uses group size 32, and NVFP4 always uses 16. For methods with block transforms (i.e., not applicable to SmoothQuant, QuaRot, and SpinQuant), the transform block size is always the same as the quantization group size.
>
> **W4: WUS in INT4.**
>
> WUS removes the Hadamard mixing and keeps only the non-orthogonal data-aware component. This can amplify certain coordinates, which is particularly harmful for INT quantization because the group scale is set by the maximum magnitude, and a larger scale creates larger quantization error. WUS is an ablation study to show that the Hadamard part is essential to spread energy more evenly across coordinates and controls the maximum magnitude. Its worse performance is expected and is not a weakness of this paper.
>
> **W5: Stochastic unbiased quantization vs RTN.**
>
> We have a discussion on this in the last paragraph of Section 6.1, and we will discuss this more explicitly in the revision. The stochastic unbiased model is used as a tractable analytical surrogate for deriving the transform. It captures the dominant scaling behavior of FP and INT quantization. Empirically, the theoretical predictions align well with the layerwise results in Table 1 and the end-to-end results.

---

> > ### Author Rebuttal · Reviewer_tLCu · 2026-04-01
> >
> > Most of my concerns have been addressed, and I appreciate the authors’ clarifications and additional experiments. I would like to increase my score.

---

### Official Review · Reviewer_piJy · 2026-03-04

**Soundness:** 4
**Presentation:** 4
**Significance:** 3
**Originality:** 3
**Overall Recommendation:** 5
**Confidence:** 4

**Summary:**

This paper reformulates the optimal transform problem from a probabilistic perspective, and then introduces WUSH, a data-aware blockwise transform for LLM weight-activation quantization. WUSH provides a mathematically optimal linear transform for floating-point formats and a near-optimal one for integer formats. The authors also designed an efficient fused GPU kernel, demonstrating accuracy improvements over existing methods on Llama3 and Qwen3 models in 4-bit.

**Compliance With Llm Reviewing Policy:**

Affirmed.

**Final Justification:**

This paper introduces WUSH, a data-aware blockwise transform for LLM weight-activation quantization that provides optimal or near-optimal linear transforms, supported by an efficient fused GPU kernel.

I thank the authors for their detailed rebuttal, which successfully addressed my primary concerns regarding computational and memory overhead, calibration dataset sensitivity, and empirical validation. In particular, the newly included complexity analysis and the sensitivity evaluation across various calibration datasets significantly strengthen the method's practical value and demonstrate its robustness for real-world deployment.

Overall, the rebuttal further reinforces the fundamental strengths of this work, and I am happy to recommend an accept.

**Key Questions For Authors:**

- Can the authors add a complexity analysis of the computational and memory costs for the offline preprocessing stage?
- Can the authors add experiments to report the required GPU memory and runtime of WUSH, compared with existing transform methods such as SpinQuant and FlatQuant?
- Given its reliance on second-order statistics, how sensitive is the WUSH transformation to the diversity of the calibration dataset?

**Limitations:**

yes

**Strengths And Weaknesses:**

Strengths:
- Mathematical proofs guarantee optimality for FP and near-optimality for INT quantizers.
- Well-structured and well-written, covering clearly visualized theory, experimental merits, and GPU kernel implementation.
- Proposes a data-aware blockwise transform, which does not require training or fine-tuning the entire model.
- Resolves ambiguity around why Hadamard rotations work and improves the MXFP4 format's competitiveness against NVFP4.

Weaknesses:
- High computational and memory cost during offline preprocessing, due to per-block SVD/Cholesky decompositions and specialized transposed memory layouts.
- The benchmark is rather limited, it would be better to evaluate more models and also report perplexity results.

---

> ### Author Rebuttal · Authors · 2026-03-30
>
> Thank you for your time and effort in reviewing this submission. We appreciate your detailed and thoughtful feedback and address your concerns and questions below.
>
> **W1 & Q1 & Q2: Computational and memory costs.**
>
> The offline processing cost of WUSH is very similar to that of the standard GPTQ algorithm because WUSH and GPTQ require the exact same activation Hessian information, and ***the cost is dominated by forwarding the calibration activations through the model***. In particular, ***when combining WUSH and GPTQ, WUSH only adds a negligible overhead on top of GPTQ***.
>
> *Complexity Analysis for Offline Preprocessing*
>
> We will add to the revision the following complexity analysis of computing WUSH for a single layer (not counting the calibration forwarding). We consider a layer with input size $d\_{\mathrm{in}}$, output size $d\_{\mathrm{out}}$, calibration tokens $d\_{\mathrm{batch}}$, and transform/quantization block size $d$. In our settings, $d$ is 16 or 32, $d\_{\mathrm{in}}$ and $d\_{\mathrm{out}}$ are several thousand, and $d\_{\mathrm{batch}}$ is hundreds of thousands. $d \ll d\_{\mathrm{in}}\ll d\_{\mathrm{batch}}$ and $d\ll d\_{\mathrm{out}}\ll d\_{\mathrm{batch}}$.
>
> We only process the diagonal blocks in the transforms, and the off-diagonal blocks are not computed or stored. For a single block, computing WUSH requires the following steps:
>
> |Step|Computation|Memory|
> |-|-|-|
> |Compute the second moments in Equation 6|$O(d^2d\_{\mathrm{batch}})$|$O(dd\_{\mathrm{batch}})$|
> |Cholesky, SVD, and matrix multiplications in Equations 6-8|$O(d^3)$|$O(d^2)$|
> |Apply transforms to weights before quantization|$O(d^2d\_{\mathrm{out}})$|$O(dd\_{\mathrm{out}})$|
> |**Overall**|$O(d^2d\_{\mathrm{batch}})$|$O(dd\_{\mathrm{batch}})$|
>
> There are $\frac{d\_{\mathrm{in}}}{d}$ blocks in total. Computing WUSH for all blocks has computational complexity of $O(dd\_{\mathrm{in}}d\_{\mathrm{batch}})$ and memory complexity of $O(d\_{\mathrm{in}}d\_{\mathrm{batch}})$.
>
> In particular, the Cholesky and SVD are conducted separately on $d\times d$ tiny matrices. They are fast, and the cost scales linearly with the number of blocks.
>
> The $\frac{d\_{\mathrm{in}}}{d}$ transforms of shape $d\times d$ are arranged and stored in transposed layouts ($d\times d\times \frac{d\_{\mathrm{in}}}{d}$) as required by our GPU kernel. Only $dd\_{\mathrm{in}}$ elements are transposed and stored. This is about a hundred times smaller than the weight matrix that has $d\_{\mathrm{in}}d\_{\mathrm{out}}$ elements to be stored. We have a table for the exact storage overheads in our reply to Reviewer `tLCu` (Weakness 2).
>
> *Cost Measurements for Offline Preprocessing*
>
> We will add to the revision the following WUSH (with RTN) preprocessing measurements. Note that the offline programs are generally not optimally implemented. And there are some configuration choices that can significantly vary the measurements. For example, one can choose to offload and recompute the calibration activations to save GPU memory at the cost of increased runtime.
>
> |Model|GPU|Time [min]|Mem [GB]|
> |-|-|-|-|
> |LLama-3.2-3B-Instruct|H100|9|10|
> |LLama-3.1-8B-Instruct|H100|19|19|
> |Qwen3-8B|H100|25|17|
> |Qwen3-14B|H100|30|20|
> |Qwen3-32B|B200|38|40|
>
> In comparison, for the LLama-3-8B-level model, SpinQuant reports 30 minutes, and FlatQuant reports 54 minutes and 26 GB.
>
> *Costs for Online Inference*
>
> As for the online inference costs, the main result in Section 5 is that WUSH can be fused efficiently and runs close to the MXFP4 matmul-only (no-transform/no-quantize) baseline, so the practical runtime overhead is tiny. The memory is dominated by the KV cache and model weights, and the storage cost of the transforms is negligible as per our analysis. We are unable to fairly compare the inference cost to SpinQuant or FlatQuant because no FP4 inference kernels are implemented for these methods.
>
> **W2: Benchmark coverage.**
>
> We have conducted experiments on five recent LLMs of different families and sizes (LLama-3.2-3B-Instruct, LLama-3.1-8B-Instruct, and Qwen3-8B/14B/32B). The results can be found in Tables 1-4 and Figures 3, 6-9, some of which are in Appendix B.
>
> In Appendix B Table 4, we include KL results for Qwen3-8B, where the WUSH and GPTQ combination is the best for both MXFP4 and NVFP4. KL divergence is a conceptually similar metric to perplexity.
>
> **Q3: Sensitivity to calibration data diversity.**
>
> Our results are aggregated over multiple random seeds as indicated in Appendix B. The box plots in Figures 3, 6-9 show the standard deviations. In addition, we show the MXFP4 results for Qwen3-8B using different datasets below. We find the results consistent across runs.
>
> ||LM Eval (Avg.)|Platinum Bench (Avg.)|
> |-|-|-|
> |***RTN-WUSH***|||
> |FineWeb|74.91|93.00|
> |C4|75.57|92.89|
> |Open-Platypus|74.99|93.12|
> |***GPTQ-WUSH***|||
> |FineWeb|75.69|93.52|
> |C4|75.78|93.53|

---

> > ### Author Rebuttal · Reviewer_piJy · 2026-04-01
> >
> > I would like to thank the authors for their detailed rebuttal. My primary concerns regarding computational and memory overhead, calibration dataset sensitivity, and empirical validation have been adequately addressed.
> >
> > In particular, the newly included complexity analysis and empirical evaluation of the offline preprocessing stage significantly strengthen the practical value of WUSH. Furthermore, the sensitivity analysis across different calibration datasets is a crucial addition; it demonstrates the robustness of the method, enhancing its credibility for real-world deployment where dataset sensitivity is often a major challenge.
> >
> > Given these improvements, I am happy to maintain my score as a positive accept.

---

### Official Review · Reviewer_hhWT · 2026-03-08

**Soundness:** 3
**Presentation:** 3
**Significance:** 3
**Originality:** 3
**Overall Recommendation:** 4
**Confidence:** 3

**Summary:**

This paper proposes a near-optimal block diagonal rotation matrix for improving the performance of weight-activation LLM quantization setups. The rotation, named "WUSH," relies on input activation statistics to construct a better weight-rotation matrix than the de-facto random Hadamard transform that is widely used in quantization. A similar construction for activation rotations based on weight statistics is also proved as "XUSH." The proposed transformations outperform Hadamard transforms and learned rotations (SpinQuant) in downstream tasks under both weight-activation and weight-only quantization setups. The authors also show that the rotations can be applied efficiently in an online manner with minimal inference overhead.

**Compliance With Llm Reviewing Policy:**

Affirmed.

**Final Justification:**

The idea is interesting but the empirical results, problem scope, and evaluation coverage could be stronger. I think the gains are pretty limited in the GPTQ case, which practically is the important case since people don't really use RTN for weight PTQ (this is different for QAT but this paper does not study QAT). I will keep my score of a weak accept.

**Key Questions For Authors:**

See above.

**Limitations:**

Mostly, see above.

**Strengths And Weaknesses:**

Strengths:
1. The proposed rotation matrices are near optimal under the objective of minimizing the immediate layerwise activation error.
2. Although I did not check the math in detail, the authors provide theoretical justification for their construction under both floating point and integer quantizer assumptions. I am not actually sure why the authors felt the need to separate out their quantizer by distortion type, since past works such as QuIP and YAQA have provided error bounds for quantization with a single distortion $\sigma^2$.
3. The proposed rotation matrices improve downstream performance over RHTs and learned rotation matrices on downstream tasks.
4. The authors provide a way to integrate WUSH into GPTQ since the two methods use the same activation Hessian but must be interleaved.

Weaknesses and Questions:
1. The authors restrict their analysis to block diagonal Tx and Tw. From my understanding, there is no reason Tw needs to be block diagonal since it is applied before quantizing the weight matrix. Only Tx needs to be block diagonal so it can be applied online in a memory bound way.
2. WUSH is data aware in the sense that it uses the Hessian of x and W with respect to the L2 error of the immediate activation output, but this Hessian is taken as an expectation over data and is static over different inputs. Is there a way to adjust XUSH online during decoding depending on the input statistic?
3. WUSH tries to minimize the immediate layerwise activation error. However, works such as YAQA and GuidedQuant have shown that this objective is actually quite bad for preserving the final model output (ie minimizing the KL to the original model output). Is there a way to modify WUSH to use a better Hessian or more 2nd order information from the rest of the model?
4. The empirical evaluation tables only show downstream task performance and not perplexity or KL to the original model. KL is probably the most important metric for quantization, so the authors should add this.
5. Does the empirical benefit of WUSH come from being able to use different Tx and Tw or the actual construction? Methods that use the RHT generally apply the inverse RHT since the orthogonality cancels out, but in practice the quantization step makes this not true. If we set Tx = Tw^-1 or something like this, would WUSH still perform well?
6. Methods like QuIP and YAQA bound the error by an incoherence metric. In the weight only setting, does WUSH minimize the incoherence better than the RHT and learned methods like SpinQuant that target kurtosis?


Side note: the timing results you reported (>4x speedup when going from BF16 to FP4) seem to be from the 5090 being able to do FP4 GEMMs with FP32 accumulate 8x faster than BF16 GEMMs with FP32 accumulate. It would be better to report timing numbers on a GB200 that has a "true" 4:1 FP4:BF16 ratio since the 5090 is a "cut down" Blackwell design.

---

> ### Author Rebuttal · Authors · 2026-03-30
>
> Thank you for the review!
>
> To avoid terminology mismatch: (1) our method uses general linear transforms, not rotations (orthogonal); and (2) the activation-side transform is XVSH, not XUSH.
>
> **Why separate FP and INT quantizer models?**
>
> We separate them because the dominant distortion mechanism differs qualitatively: for groupwise FP formats, the error is approximately proportional to the magnitude of each value, while for INT with AbsMax scaling, it is controlled by the group scale / max magnitude. These lead to different optimization problems and different conclusions about when transforms help.
>
> Compared to the previous works, the QuIP paper only considers the INT format. The QuIP#/YAQA papers model the generic quantizers in a loose bound of $\mathbb{E}[(q(x)-x)^2]\le\sigma^2$, which naturally leads to results in the form of $\mathbb{E}[\mathrm{loss}]\le\text{bound}$. By contrast, we use separate FP- and INT-specific stochastic error models, which let us derive $\mathbb{E}[(q(x)-x)^2]$ and $\mathbb{E}[\mathrm{loss}]$ directly and minimize the expectation $\mathbb{E}[\mathrm{loss}]$ itself (for FP) or an asymptotically tight bound (for INT), rather than only minimizing the format-agnostic upper bounds (which are loose proxies for the actual FP/INT formats). Another key difference is that our focus is on RTN, which is practically relevant for activation quantization, while the QuIP/QuIP#/YAQA bounds are proved for LDLQ/GPTQ.
>
> **Q1 & Q5: Tw and Tx.**
>
> $\boldsymbol{T}\_\mathrm{W}$ and $\boldsymbol{T}\_\mathrm{X}$ are not two independent transforms. To ensure the equivalence of the unquantized layer output before and after transforms, i.e., $\boldsymbol{W}^\top\boldsymbol{X}=(\boldsymbol{T}\_\mathrm{W}\boldsymbol{W})^\top(\boldsymbol{T}\_\mathrm{X}\boldsymbol{X})$, we indeed have a constraint of $\boldsymbol{T}\_\mathrm{W}=\boldsymbol{T}\_\mathrm{X}^{-\top}$ as stated in Section 4.1. The WUSH and XVSH transforms satisfy this constraint by construction (Equations 8-9). Please see our reply to reviewer `j34m` (Question 2) for a short proof.
>
> Under the constraint of $\boldsymbol{T}\_\mathrm{W}=\boldsymbol{T}\_\mathrm{X}^{-\top}$, $\boldsymbol{T}\_\mathrm{W}$ must be block diagonal if $\boldsymbol{T}\_\mathrm{X}$ is block diagonal.
>
> The empirical benefit of WUSH comes from its construction derived in Section 4. In Appendix A.2, we also intuitively demonstrate that WUSH, which is non-orthogonal in general, can alleviate outliers in the spectrum space and outperform an orthogonal transform such as the Hadamard.
>
> **Q2: Adjusting XVSH online.**
>
> A per-input adaptive transform is possible in principle, but would require recomputing statistics online and would significantly complicate the memory-bound decoding path. In this paper, our goal is to keep the method as efficient as the blockwise Hadamard baseline using fused kernels. We think this direction is interesting, and plan to investigate it for future work.
>
> **Q3 & Q4: KL divergence.**
>
> In Appendix B Table 4, we provide KL results for Qwen3-8B, showing that the WUSH and GPTQ combination is also the best for both MXFP4 and NVFP4 under this metric.
>
> We agree that immediate layerwise error is only a proxy. We chose it because it is the canonical PTQ objective and, importantly, it yields a tractable closed-form transform. That said, the construction itself only needs suitable positive-definite second-order matrices, so in principle one could replace the current moments/Hessian surrogate with downstream-aware curvature (empirical Fisher) or KL-guided surrogates. We view this as a promising extension in the future.
>
> **Q6: Incoherence.**
>
> Blockwise, WUSH also provides the optimal incoherence ($\mu=1$). The post-transform blockwise Hessian (up to a constant factor) is $\boldsymbol{T}\_{\mathrm{wush}}\boldsymbol{X}'\boldsymbol{X}'^\top\boldsymbol{T}\_{\mathrm{wush}}^\top$$=\boldsymbol{H}\boldsymbol{S}^{-\frac{1}{2}}\boldsymbol{U}^\top(\boldsymbol{W}'^\top\boldsymbol{X}')(\boldsymbol{W}'^\top\boldsymbol{X}')^\top\boldsymbol{U}\boldsymbol{S}^{-\frac{1}{2}}\boldsymbol{H}^\top$$=\boldsymbol{H}\boldsymbol{S}^{-\frac{1}{2}}\boldsymbol{U}^\top(\boldsymbol{U}\boldsymbol{S}\boldsymbol{V}^\top)(\boldsymbol{U}\boldsymbol{S}\boldsymbol{V}^\top)^\top\boldsymbol{U}\boldsymbol{S}^{-\frac{1}{2}}\boldsymbol{H}^\top$$=\boldsymbol{H}\boldsymbol{S}\boldsymbol{H}^\top$. The eigenbasis of the Hessian is a normalized Hadamard matrix whose elements are $\pm \frac{1}{\sqrt{d}}$. The incoherence factor $\mu$ is defined as $\sqrt{d}$ times the maximum absolute value of the eigenbasis matrix, so $\mu=1$ in this case, which is also the minimum possible incoherence by its definition.
>
> **Timing on GPU.**
>
> We agree with this observation. Our intent is to demonstrate that online WUSH is practical on currently available Blackwell-class consumer hardware. We are working on optimizing our kernels for more hardware, but this is not the focus of the current paper.

---

> > ### Author Rebuttal · Reviewer_hhWT · 2026-04-02
> >
> > Thanks for the response and clarifications. I will keep my current score. I think the method presented in this paper is useful and the scale of the contribution is in line with prior transformation-based works like SpinQuant, but this entire subfield of optimizing transformations has largely been incremental.
> >
> > The main technical reservation I have regarding this method is that it, like other "function-preserving transformation" papers, is offline and static for inputs. I am guessing that this is why, despite how much better WUSH should be than the RHT, GPTQ-WUSH is only slightly better than GPTQ-H (MR-GPTQ) in Table 2. It should be possible to collect online statistics from the current activation/sequence and apply a modified $T_x \neq T_w^{-T}$ in a memory bound way, but I acknowledge the authors have decided to leave it for future work.
> >
> > Finally, is there a reason why you did not compare to FlatQuant? FlatQuant does a lot better than SpinQuant and doesn't cost that much more. You have SpinQuant in your tables.

---

> > > ### Author Response · Authors · 2026-04-06
> > >
> > > Thank you for the follow-up! We address your new concerns and questions below.
> > >
> > > **Scale of contribution.**
> > >
> > > We believe WUSH goes beyond incremental. WUSH is near-optimal under some standard assumptions. WUSH adapts to the data distribution in a closed-form, without requiring iterative tuning or grid search over parameters. The closed-form construction enables us to provide a rigorous theoretical explanation of why the popular Hadamard transform (QuIP#, QuaRot, etc.) helps and why a *non-orthogonal, data-aware* construction is even better. In that sense, our contribution is not only a new transform but also a new theoretical framework for the class of transform-based methods.
> > >
> > > More broadly, transform optimization is now recognized as a core part of the LLM quantization pipeline. Recent review work [1] explicitly decomposes quantization into pre-quantization transformation and error mitigation, and reports that optimized rotation/scaling is the strongest transformation strategy among existing options.
> > >
> > > **Online adaptive transforms.**
> > >
> > > The goal of this paper is to improve accuracy at essentially no additional runtime cost over the current state-of-the-art deployable baseline, GPTQ-H (aka MR-GPTQ [2] and BRQ [3]). For that reason, we focus on offline-computed static transforms, which can be calibrated once and then applied efficiently at inference time.
> > >
> > > By contrast, the online-adjusted dynamic transform you suggest is very interesting, but substantially more complex. For decoder-only LLMs, online statistics are only meaningful at the current-token level: future activations are unavailable, past ones cannot be changed, and batch-level statistics are problematic for privacy reasons. Therefore, a truly online adaptive method would need a separate $d\times d$ transform for each $d$-dimensional quantization group, which may require $d$ times more memory than the activations themselves. More fundamentally, once a full $d\times d$ matrix is allowed to vary online per group, the problem is no longer just finding a better transform before RTN, but becomes much closer to designing a new quantization algorithm altogether. We therefore consider your proposal as a very interesting direction for future work, rather than an extension of the current paper.
> > >
> > > **GPTQ-WUSH vs GPTQ-H.**
> > >
> > > Our interpretation is that GPTQ-H is a very strong baseline for the model and quantization format in Table 2, so the remaining room for improvement is small. However, for other models and quantization formats where GPTQ-H has a larger gap to the full-precision accuracy, the GPTQ-WUSH improvement can be more significant, such as for the Llama-3.2-3B-Instruct and MXFP4 in Figure 8.
> > >
> > > **FlatQuant comparison.**
> > >
> > > We focused our empirical evaluation on the recently proposed MXFP4 and NVFP4 formats, since our theory predicts that WUSH is especially effective for FP quantization, and these formats are the only ones with broad support on modern GPUs. The strongest baseline for FP quantization is GPTQ-H (MR-GPTQ/BRQ), so our experiments are prioritized around this method. By contrast, FlatQuant is designed for INT quantization, and its storage format is not naturally suited to building a fused transform-quantization kernel that achieves high utilization on current FP-oriented hardware. Therefore, it was not our main evaluation target.
> > >
> > > That said, we agree that FlatQuant is a relevant point of reference. We have indeed considered including it by modifying their code to support FP types and running the evaluations ourselves. However, the versions of the public code we tested before our submission contained issues that we were unable to resolve in time for the ICML initial submission.
> > >
> > > We have now managed to obtain preliminary results for FlatQuant for MXFP4 and RTN. Compared to the numbers in Table 2, RTN-WUSH is still better on average.
> > >
> > > ||MMLU-CoT|GSM8K|HellaSwag|WinoGrande|Average|Recovery|
> > > |-|-|-|-|-|-|-|
> > > |RTN-FlatQuant|65.97|70.58|77.94|75.14|72.41|91.74|
> > > |RTN-WUSH|66.85|75.16|77.28|73.56|**73.21**|**92.75**|
> > >
> > > ---
> > >
> > > **References**
> > >
> > > [1] Liu et al. A Comprehensive Evaluation on Quantization Techniques for Large Language Models. arXiv 2026. URL https://arxiv.org/abs/2507.17417
> > >
> > > [2] Egiazarian et al. Bridging the Gap Between Promise and Performance for Microscaling FP4 Quantization. ICLR 2026. URL https://openreview.net/forum?id=zCBGe9AqJZ
> > >
> > > [3] Shao et al. Block Rotation is All You Need for MXFP4 Quantization. arXiv 2025. URL https://arxiv.org/abs/2511.04214

---

### Official Review · Reviewer_j34m · 2026-03-09

**Soundness:** 3
**Presentation:** 2
**Significance:** 3
**Originality:** 4
**Overall Recommendation:** 4
**Confidence:** 4

**Summary:**

This paper focuses on W4A4 PTQ for LLMs, proposing a transformation-based method named WUSH.
The paper states that prior methods, such as the Hadamard transform, rely on fixed and data-agnostic transforms, which are not theoretically optimal. WUSH addresses this issue by integrating a Hadamard backbone with an adaptive, data-dependent component derived from the second-order statistics of calibration data. This results in a non-orthogonal transform tailored to the actual data distribution.
The authors claim that WUSH is theoretically near-optimal for both floating-point and integer block quantizers, supporting this with extensive mathematical proofs.
While WUSH involves online transforms on activations, the authors implement specialized GPU kernels to ensure high efficiency.
Empirically, WUSH demonstrates accuracy enhancements on standard benchmarks, along with significant speedups over standard BF16 inference.

**Compliance With Llm Reviewing Policy:**

Affirmed.

**Final Justification:**

Authors have clarified most of my main concerns. I maintain my positive assessment.

**Key Questions For Authors:**

- authors says "As opposed to prior methods such as SpinQuant or FlatQuant , our approach is closed-form and calibration-based, as it does not require training or fine-tuning the entire model." Actually, SpinQuant and FlatQuant are also calibration-based, and only fine-tune quantization parameters.
- how to ensure equivalent after involving Tw and Tx?
- comparing to pure quantizaiton (without online calculations), what is the inference latency WUSH achieve?

**Limitations:**

yes

**Strengths And Weaknesses:**

### Strengths
- provides theoretical derivations for closed-form transforms, rather than just relying on heuristics.
- implements GPU kernels to make the proposed online transforms practical for inference.
- shows empirical accuracy improvements on standard benchmarks and reports hardware speedups over BF16.

### Weaknesses
- the paper is not easy to follow. The introduction should be reorganized, for instance, by moving the detailed formulation explanation to the methodology section.
- the caption for Figure 1 is too long and dense, making it difficult to follow.
- for Figure 2, the authors need to clarify why the kernel without transform or quantization (None) achieves a ~6x speedup versus BF16.
- there is a lack of comprehensive ablation studies.

---

> ### Author Rebuttal · Authors · 2026-03-30
>
> Thank you for your time and dedication to reviewing this submission. We address your concerns and questions below.
>
> **W1 & W2: Presentation of the introduction section and the Figure 1 caption.**
>
> Thank you for the feedback. We will reorganize the introduction and make the Figure 1 caption shorter and cleaner in the revision.
>
> **W3: Figure 2 baseline speedup.**
>
> "None" denotes the pure MXFP4 GEMM kernel without transform or quantization overhead, so its speedup over BF16 reflects the low-precision matmul advantage itself, not our transform. The theoretical peak speedup of 8x is for infinitely large matrices. The ~6x is a practical peak speedup for the shapes of real matrices in our test models. For details on Figure 2, please see the last paragraph of Section 5.
>
> **W4: Ablation studies.**
>
> We have a broad set of ablation studies in the paper, though they are not explicitly labeled using the word "ablation". The current results already compare different transform types (identity I, random rotation R, Hadamard H, WUS, WUSH) and quantization methods (RTN, GPTQ, etc.). The empirical evaluation results can be found in Tables 1-4 and Figures 3, 6-9. In particular, H vs WUS vs WUSH separates the effects of the Hadamard backbone and the adaptive non-orthogonal component. We also theoretically discussed the necessity of each non-trivial component of WUSH in Sections 4.2.3, 4.3.3, and Figure 1. In addition, we add the results on different calibration datasets in our reply to Reviewer `piJy` (Question 3), showing WUSH construction consistently yields good results under the dataset diversity.
>
> **Q1: SpinQuant/FlatQuant statement.**
>
> Our intended distinction was not calibration-based vs non-calibration-based, but rather closed-form vs learned or iteratively optimized (e.g., gradient descent). To avoid confusion, we will revise this sentence to say that WUSH is a closed-form calibration-driven transform, whereas SpinQuant/FlatQuant learn transforms on calibration data.
>
> **Q2: Equivalence after involving Tw and Tx.**
>
> The equivalence of $\boldsymbol{W}^{\top} \boldsymbol{X} = (\boldsymbol{T}\_\mathrm{W} \boldsymbol{W})^{\top} (\boldsymbol{T}\_\mathrm{X} \boldsymbol{X})$ is guaranteed if $\boldsymbol{T}\_\mathrm{W} = \boldsymbol{T}\_\mathrm{X}^{-\top}$, i.e., $\boldsymbol{T}\_\mathrm{W}^{\top} \boldsymbol{T}\_\mathrm{X} = \mathbf{I}$, which implies $(\boldsymbol{T}\_\mathrm{W} \boldsymbol{W})^{\top} (\boldsymbol{T}\_\mathrm{X} \boldsymbol{X}) = \boldsymbol{W}^{\top} \boldsymbol{T}\_\mathrm{W}^{\top} \boldsymbol{T}\_\mathrm{X} \boldsymbol{X} = \boldsymbol{W}^{\top} \boldsymbol{X}$.
>
> By Equations 8-9, for a block (we omit the block index $(i)$ in the subscripts here), the constructions of the activation-side transform WUSH $\boldsymbol{T}\_\mathrm{X} = \boldsymbol{T}\_\mathrm{wush} = \boldsymbol{H} \boldsymbol{S}^{-\frac{1}{2}} \boldsymbol{U}^{\top} \boldsymbol{W}'^{\top}$ and the weight-side counterpart XVSH  $\boldsymbol{T}\_\mathrm{W} = \boldsymbol{T}\_\mathrm{xvsh} = \boldsymbol{H} \boldsymbol{S}^{-\frac{1}{2}} \boldsymbol{V}^{\top} \boldsymbol{X}'^{\top}$ exactly satisfy this condition:
>
> $\boldsymbol{T}\_\mathrm{W}^{\top} \boldsymbol{T}\_\mathrm{X}$
> $=(\boldsymbol{H} \boldsymbol{S}^{-\frac{1}{2}} \boldsymbol{V}^{\top} \boldsymbol{X}'^{\top})^{\top} (\boldsymbol{H} \boldsymbol{S}^{-\frac{1}{2}} \boldsymbol{U}^{\top} \boldsymbol{W}'^{\top})$
> $= \boldsymbol{X}' \boldsymbol{V} \boldsymbol{S}^{-\frac{1}{2}} \boldsymbol{H}^{\top} \boldsymbol{H} \boldsymbol{S}^{-\frac{1}{2}} \boldsymbol{U}^{\top} \boldsymbol{W}'^{\top}$
> $= \boldsymbol{X}' \boldsymbol{V} \boldsymbol{S}^{-\frac{1}{2}} \boldsymbol{S}^{-\frac{1}{2}} \boldsymbol{U}^{\top} \boldsymbol{W}'^{\top}$
> $= \boldsymbol{X}' \boldsymbol{V} \boldsymbol{S}^{-1} \boldsymbol{U}^{\top} \boldsymbol{W}'^{\top}$
> $= \boldsymbol{X}' (\boldsymbol{U} \boldsymbol{S} \boldsymbol{V}^{\top})^{-1} \boldsymbol{W}'^{\top}$
> $= \boldsymbol{X}' (\boldsymbol{W}'^{\top} \boldsymbol{X}')^{-1} \boldsymbol{W}'^{\top}$
> $= \boldsymbol{X}' \boldsymbol{X}'^{-1} \boldsymbol{W}'^{-\top} \boldsymbol{W}'^{\top}$
> $= \mathbf{I}$
> .
>
> Therefore, the transform itself preserves the exact unquantized layer output and only the subsequent quantization introduces approximation error.
>
> **Q3: Inference latency relative to pure quantization.**
>
> Based on Figure 2, the speedup should be between the matmul-only ("None") setting and the fused Hadamard-quantize-matmul ("H + Quant") setting. Given that the WUSH transform only yields a small gap to the matmul-only upper bound, the inference latency of WUSH should be very close to that of the pure quantization setting. However, we did not include a fused kernel for the MXFP4 matmul and the pure quantization without transform because this combination does not yield good accuracy, as shown in our evaluation results. Creating such a kernel would require a large amount of effort, and the resulting model would have poor accuracy.

---

> > ### Author Rebuttal · Reviewer_j34m · 2026-04-03
> >
> > Thank you for the detailed rebuttal. Authors have clarified most of my main concerns. But I still have concerns about the completeness and presentation of the ablation studies, and the lack of direct latency comparison. Overall, I maintain my positive assessment.

---

> > > ### Author Response · Authors · 2026-04-06
> > >
> > > Thank you for the follow-up!
> > >
> > > In the revision, we will make the current component-wise comparisons more explicit as ablation studies. If there are particular ablations you feel are still missing beyond these, we would appreciate specific suggestions.
> > >
> > > We will also state our latency analysis more explicitly in the revision.

---

### Decision · Program_Chairs · 2026-04-30

**Decision:**

Accept (regular)

**Comment:**

Having read the reviews and rebuttal carefully, I find that the submission makes a sufficiently strong contribution to warrant acceptance. The discussion helped clarify several points, and the remaining issues are not significant enough to affect the overall recommendation.